# Multi-Qubit Bose–Einstein Condensate Trap for Atomic Boson Sampling

**DOI:** 10.3390/e24121771

**Published:** 2022-12-03

**Authors:** Sergey Tarasov, William Shannon, Vladimir Kocharovsky, Vitaly Kocharovsky

**Affiliations:** 1Institute of Applied Physics, Russian Academy of Sciences, Nizhny Novgorod 603950, Russia; 2Department of Physics and Astronomy and Institute for Quantum Science and Engineering, Texas A&M University, College Station, TX 77843-4242, USA

**Keywords:** Bose–Einstein condensation, Gaussian boson sampling, quantum advantage, NP-hard problem

## Abstract

We propose a multi-qubit Bose–Einstein-condensate (BEC) trap as a platform for studies of quantum statistical phenomena in many-body interacting systems. In particular, it could facilitate testing atomic boson sampling of the excited-state occupations and its quantum advantage over classical computing in a full, controllable and clear way. Contrary to a linear interferometer enabling Gaussian boson sampling of non-interacting non-equilibrium photons, the BEC trap platform pertains to an interacting equilibrium many-body system of atoms. We discuss a basic model and the main features of such a multi-qubit BEC trap.

## 1. Introduction to Quantum Statistical Physics of Atomic Boson Sampling in a BEC Trap

### 1.1. The Essence of the Problem

Recently, a stationary stochastic process of many-body fluctuations of the excited-atom occupations in a trapped Bose–Einstein-condensed gas has been suggested for quantum simulation of the ♯P-hard problem of boson sampling [1]. Such an atomic boson sampling, based on the Bose–Einstein-condensate (BEC) platform, is an alternative to a well-known photonic boson sampling based on the linear interferometer platform [2,3,4,5,6,7,8,9,10,11,12,13,14,15,16,17,18,19,20,21,22,23,24,25]. It has the potential to demonstrate quantum advantage [26,27,28,29,30] of the many-body interacting systems over classical computers. For a full and clear demonstration of a ♯P-hardness of computing atom-excitation sampling, a condensate should be nonuniformly spread over an entire BEC trap and provide, via an interparticle interaction, multimode Bogoliubov coupling between a large number of excited atom states. Moreover, all of the above parameters of the many-body system should be controllable in a wide range to ensure sufficient variability of the observed joint occupation statistics of the excited states or coarse-grained groups of excited states. So, there is an open problem of designing BEC traps most suitable for experimental studies of various phenomena associated with atomic boson sampling.

The present paper is devoted to this problem: We discuss a basic model of a potential design of the multi-qubit BEC trap that could provide the required conditions and be particularly suitable for atomic-boson-sampling experiments. It is inspired by an analogy with multi-qubit or multi-qudit systems [31,32] and could look like a system of a finite number, *Q*, of single-qubit or -qudit cells shown in Figure 1 in a two-dimensional (2D) case.

Remarkably, a direct measurement of fluctuations in a total occupation of the noncondensate in cold dilute gases has already been achieved [33,34]. Splitting the noncondensate into some parts associated with the groups of excited states and measuring atom-number fluctuations in the occupations of those parts is the next important step in the many-body statistical physics toward testing quantum advantage. It is beyond the bulk of previous studies of the BEC phenomena, which is devoted to the mean properties of the condensate and quasiparticles, and it is also beyond the previous studies of fluctuations in the total occupation of the condensate (see, e.g., [34,35,36,37,38,39,40,41,42]). The atom-number fluctuations are especially important for the applications related to quantum information science and matter-wave interferometers [43], including Ramsey [44,45] and Mach–Zehnder [46] on-chip interferometers. In the literature, there are also other interesting discussions of the atom-number fluctuations associated with a subvolume of a BEC trap [47,48], BEC collapse [49], and squeezed states [44].

### 1.2. What Is the Atomic Boson Sampling?

In statistical theory, sampling is a selection of events (subsets) from within a sample space of all possible outcomes (or results or sample points) to mimic the characteristics of the probability distribution in a probability space (a probabilistic model). Atomic boson sampling means sampling from the excited-state occupations of identical Bose atoms subject to interparticle scattering (interaction) in a trapped Bose–Einstein condensed gas within a statistical ensemble of a given experimental setup. The atoms forming the condensate are not counted. One can consider the integer occupations $n_{k}=0,1,2,\ldots$ and their joint probability distribution $\rho (\left\{n_{k}\right\})$ for the individual orthonormal excited states $\left\{\phi_{k}\left(r\right)|\;k=1,2,\ldots \right\}$, orthogonal to the condensate wave function, or for groups of such excited states. The simultaneous measurement of their occupations has to be completed by multiple detectors via projecting atoms onto preselected subsets (groups) of the excited states. The latter subsets determine the sampling probability distribution in question.

Condensed-matter statistical physics of a mesoscopic system of *N* atoms confined in a trap is highly nontrivial due to an interaction between massive atoms taking place on a background of the Bose–Einstein condensate formed by the same interacting atoms via spontaneous symmetry breaking at a critical temperature $T_{c}$. Quantum many-body fluctuations in this system remain ♯P-hard for computing [1] even in equilibrium and even within the grand-canonical-ensemble [50] and Bogoliubov–Popov approximations [51,52,53]. For simplicity’s sake, we adopt these approximations in the present paper and assume that the temperature is well below the critical region of the BEC phase transition, $T\ll T_{c}$.

The computational ♯P-hardness of atomic boson sampling is a real property of the interacting BEC gas, not just a feature of the Bogoliubov–Popov approximation. It follows from the exact non-perturbative theory of critical fluctuations in BEC, which is based on the non-polynomial diagram technique [54,55,56] and also leads to the representation of the joint probability distribution of the occupations of the bare excited atomic states in terms of the hafnian of a matrix associated with a correlation matrix which is similar to the one given by the stated approximation. In other words, the ♯P-hardness is a robust property of atomic boson sampling in the sense that it manifests itself (survives) even in the Bogoliubov–Popov, i.e., the first order with respect to the interaction parameter, mean-field approximation.

Each act of atomic boson sampling occurs by means of a measurement of the occupations of excited states or groups of them. One can employ, say, a simultaneous optical multi-detector imaging. Then, the system of interacting atoms returns back to its equilibrium state and becomes ready for the next act of sampling/measurement. In a sense, the system of atoms resets itself. This is true both if the atoms were reloaded into the trap after being released from the trap in the case of a trap-destruction measurement or if the atoms were not removed from the trap. Thus, the atomic boson sampler is not a quantum simulator of some input signal or some artificial, controlled process. The system of atoms in the BEC trap equipped with the atom-number detectors is just a quantum generator of random strings of atom-excitation occupations based on the natural process of persistent equilibrium fluctuations.

Surprisingly, atomic boson sampling has the potential to demonstrate a quantum advantage that is similar to the one of Gaussian boson sampling of non-interacting photons in a linear interferometer.

### 1.3. Comparison with Photonic Boson Sampling in a Linear Interferometer

Physics of photonic boson sampling in a linear interferometer, widely studied in the past decade [2,3,4,5,6,7,8,9,10,11,12,13,14,15,16,17,18,19,20,21,22,23,24,25], looks significantly simpler since photons are non-interacting, massless and enter the interferometer in a given quantum (e.g., Fock or squeezed) state from the external sophisticated on-demand light sources. Importantly, atomic boson sampling from an equilibrium BEC trap does not require external sources of atoms in a prescribed quantum state because the interacting atoms generate squeezing by themselves. Thus, in order to demonstrate the quantum advantage in a linear interferometer, one has to prepare the system of photons in a prescribed nonequilibrium state, whereas a usual thermal state of atoms in the BEC trap is enough in the case of atomic boson sampling. The reason for switching from the original proposal [3] to the Gaussian boson sampling protocol was an absence of the on-demand single-photon sources and availability of the reliable on-demand sources of input photons in the Gaussian/squeezed states based on the process of a parametric down-conversion [4,5,23].

The computational ♯P-hardness of the joint excited-state occupation fluctuations in the BEC trap exists by itself and does not require an adjustment or fine tuning of any input, processing, interaction or coupling parameters. On the contrary, Gaussian boson sampling in a linear interferometer, like other usually discussed nonequilibrium quantum processors and simulators, requires a lossless propagation through a system of beam couplers, splitters, and phase shifters as well as external sources of photons in squeezed states. Moreover, the main limiting factor for a demonstration of quantum advantage in photonic sampling is an exponential growth of photon losses on the input–output propagation with an increasing number of optical channels and coupling elements. Such a limiting factor is absent in the case of atomic boson sampling.

### 1.4. Why Atomic Boson Sampling Is ♯P-Hard for Computing?—A Brief Theory

Despite a number of outstanding differences stated above, both atomic and photonic boson samplings belong to the same ♯P-hard complexity class. The fact is that in both cases, the sampling (i.e., joint boson-occupation) probability distribution is determined by a hafnian (or, sometimes, a permanent) of matrices built from an appropriate covariance matrix *G* of the boson creation and annihilation operators. The above fact follows from the analytical calculation of the characteristic function (that is, Fourier transform) of the joint occupation probability distribution by means of the Wigner transform technique and application of the Hafnian Master Theorem that gives an explicit Taylor expansion of this characteristic function via the aforementioned hafnian [1,57]. It is well known that for a general n×n matrix, computing the hafnian or permanent is ♯P-hard [58,59,60], that is, requires an exponentially large number of operations $∼n^{3}2^{n/2}$ or $∼n^{2}2^{n}$, respectively, for the best general-purpose algorithms [60,61,62,63]. Contrary to a determinant that can be computed in polynomial time $∼n^{3}$ via Gaussian elimination, the hafnian and permanent matrix functions are intimately related to the analysis of the ♯P-hard problems [26,64].

Yet, the computational ♯P-hardness (which is the basis of quantum advantage) of atomic boson sampling pertains only if the matrices under the hafnian and the covariance matrix *G* from which they are built are variable/controllable in a wide range by varying the trapping potential, interparticle interaction (via Feshbach resonance [65]), temperature, number of trapped atoms and other parameters of the BEC trap [66]. In other words, those matrices should not be restricted to a narrow subset of matrices for which the hafnian could be computed in polynomial time by some efficient approximation or algorithm, such as a fully polynomial random approximation scheme (FPRAS) [59], a recurrence method [67], etc.

In order to understand how to satisfy this requirement in the design of the BEC trap, we need to know how the covariance matrix *G* depends on the trap parameters. Fortunately, we have an analytical formula for the *G* via the matrix *R* of the Bogoliubov transformation,
(1)$$G=RDR^{†}+\frac{1}{2}\left(RR^{†}-1\right);\;\;G\equiv \langle \;\;:\;\left(\begin{array}{c} \;{\hat{a}}^{†}\\ \hat{a} \end{array}\right){\left(\begin{array}{c} \;{\hat{a}}^{†}\\ \hat{a} \end{array}\right)}^{T}\;\;:\;\;\rangle ,\;\;D=\left[\begin{array}{cc} D_{1} & 0\\ 0 & D_{1} \end{array}\right],\;\;D_{1}=\underset{j}{⨁}\frac{1}{e^{E_{j}/T}-1}.$$
Here, the boldface operator vectors ${\hat{a}}^{†}={\left({\hat{a}}_{1}^{†},{\hat{a}}_{2}^{†},\ldots \right)}^{T}$ and $\hat{a}={\left({\hat{a}}_{1},{\hat{a}}_{2},\ldots \right)}^{T}$ are the column-vectors of the creation and annihilation operators for bare excited atom states, respectively. The superscripts ${}^{T}$ and ${}^{†}$ denote transpose and conjugate transpose, respectively. The angles stand for the statistical average. The colons denote the normal ordering of operators, meaning that all creation operators stand to the left from the annihilation ones. The 2×2 block-diagonal matrix *D* include the Bose–Einstein thermal occupation numbers for the quasiparticles of eigenenergies $\left\{E_{j}\right\}$. The matrix *R* performs the Bogoliubov transformation
(2)$$\left(\begin{array}{c} \;{\hat{a}}^{†}\\ \hat{a} \end{array}\right)=R\left(\begin{array}{c} \;{\hat{b}}^{†}\\ \hat{b} \end{array}\right),\;R=\left[\begin{array}{cc} U^{*} & 0\\ 0 & U \end{array}\right]\left[\begin{array}{cc} \cosh\;r^{*} & {\left(e^{i\theta }\sinh\;r\right)}^{*}\\ e^{i\theta }\sinh\;r & \cosh\;r \end{array}\right],$$
from the representation of the excited-particle field operator in terms of the quasiparticle creation and annihilation operators ${\hat{b}}^{†}={\left({\hat{b}}_{1}^{†},{\hat{b}}_{2}^{†},\ldots \right)}^{T}$ and $\hat{b}={\left({\hat{b}}_{1},{\hat{b}}_{2},\ldots \right)}^{T}$ to the representation in terms of the bare-particle creation and annihilation operators, that is,
(3)$${\hat{\psi }}_{\mathrm{ex}}\left(r\right)=\sum_{k\neq 0}\phi_{k}\left(r\right){\hat{a}}_{k}=\sum_{j}\left(u_{j}\left(r\right){\hat{b}}_{j}+v_{j}^{*}\left(r\right){\hat{b}}_{j}^{†}\right).$$

Within the mean-field Bogoliubov–Popov approximation [51,52,53], adopted in the present paper, a Bose–Einstein-condensed gas is described via the Hamiltonian given by a quadratic form of the bare-particle creation and annihilation operators
(4)$$\begin{array}{cc} \; & \hat{H}=\sum_{k,k^{\prime }}\left({\hat{a}}_{k}^{†}{\hat{a}}_{k^{\prime }}+\frac{1}{2}\right)\int \phi_{k}^{*}\left(-\frac{\hbar^{2}\Delta }{2m}+U\left(r\right)-\mu +2g\left(N_{0}{\left|\phi_{0}\left(r\right)\right|}^{2}+n_{\mathrm{ex}}\left(r\right)\right)\right)\phi_{k^{\prime }}d^{3}r\\ \; & +\frac{gN_{0}}{2}\sum_{k,k^{\prime }}{\hat{a}}_{k}^{†}{\hat{a}}_{k^{\prime }}^{†}\int \phi_{k}^{*}\phi_{0}^{2}\phi_{k^{\prime }}^{*}\;d^{3}r\;+\;\frac{gN_{0}}{2}\sum_{k,k^{\prime }}{\hat{a}}_{k}{\hat{a}}_{k^{\prime }}\int \phi_{k}{\left(\phi_{0}^{*}\right)}^{2}\phi_{k^{\prime }}\;d^{3}r\;. \end{array}$$
Here, Δ is the three-dimensional Laplace operator, *m* is a particle mass, U(r) is an external potential, $g=4\pi \hbar^{2}a/m$ is an interaction constant, *a* is a *s*-scattering length, μ is a chemical potential, $N_{0}$ is the mean number of particles in the condensate, $n_{\mathrm{ex}}\left(r\right)$ is the mean density of the excited particle fraction (the noncondensate), and $\phi_{0}\left(r\right)$ is a condensate wave function normalized to unity, $\int_{V}\left|\phi_{0}^{2}\right|d^{3}r=1$. The superscript * means complex conjugation. The Hamilton operator (Equation 4) can be equivalently rewritten in the matrix form via the (2×2)-block Hamiltonian matrix *H* as follows
(5)$$\hat{H}={\left(\begin{array}{c} {\hat{a}}^{†}\\ \hat{a} \end{array}\right)}^{T}\;H\;\left(\begin{array}{c} {\hat{a}}^{†}\\ \hat{a} \end{array}\right),\;H=\left[\begin{array}{cc} \tilde{K} & K\\ K^{*} & {\tilde{K}}^{*} \end{array}\right].$$
The Bogoliubov transformation (Equation 2) diagonalizes the blocks *K* responsible for the co-rotating contributions ${\hat{a}}_{k}^{†}{\hat{a}}_{k^{\prime }}$ or ${\hat{a}}_{k}{\hat{a}}_{k^{\prime }}^{†}$ to the Hamiltonian and nullifies the blocks $\tilde{K}$ responsible for the counter-rotating contributions ${\hat{a}}_{k}^{†}{\hat{a}}_{k^{\prime }}^{†}$ or ${\hat{a}}_{k}{\hat{a}}_{k^{\prime }}$ to the Hamiltonian,
(6)$$R^{T}HR=\left[\begin{array}{cc} 0 & E\\ E & 0 \end{array}\right],\;E=\mathrm{diag}\left\{E_{j}|\;j=1,2,\ldots \right\}.$$
The point is that the quasiparticles are the eigenstates of the Bogoliubov Hamiltonian with the eigenenergies $\left\{E_{j}|\;j=1,2,\ldots \right\}$.

The multimode squeezing matrix [1,68,69,70,71,72,73] $r=(r_{k,k^{\prime }})$, which is a positive semi-definite Hermitian matrix, as well as the unitary matrices *U* and $e^{i\theta }$, are calculated in [1]. Any additional unitary transformation *V* to another complete orthonormal set of excited states $\left\{\phi_{k}^{\prime }\left(r\right)|\;k=1,2,\ldots \right\}$ in the single-particle Hilbert space,
(7)$$\phi_{k}=\sum_{k^{\prime }\neq 0}V_{k^{\prime },k}\phi_{k^{\prime }}^{\prime },$$
results in the Bogoliubov transformation $R^{\prime }$ which differs from the *R* in Equation (Equation 2) just by replacement of the unitary *U* with the composite unitary transformation $U^{\prime }=VU$.

Clearly, the Bogoliubov transformation (Equation 2) is a superposition of the squeezing and unitary transformations. In essence, their matrices *r* and *U* determine the complexity and variability of the covariance matrix *G*, since the classical thermal occupations of quasiparticles entering the matrix *D* are easy to compute. As a result, the ultimate reason for the ♯P-hardness (quantum advantage) of atomic boson sampling is an interplay between the squeezing (found in [74]) due to the interparticle interactions and interference due to the unitary mixing of bare-particle excited states. If either the squeezing or interference vanishes, then the joint occupation probabilities can be computed classically in polynomial time.

### 1.5. The Content of the Paper

Based on the main aspects of a truly hard for computing and innovative problem of atomic boson sampling in a BEC trap explained in the Introduction (Section 1), we formulate, in Section 2, general requirements for the BEC trap design facilitating testing the quantum advantage of atomic boson sampling in a full, controllable and clear way. A basic model of a multi-qubit BEC trap is devised in Section 3. In Section 4 and Section 5, we present the analytical and numerical results for the single-particle energy spectrum and eigenstates in the one- and two-dimensional multi-qubit traps, respectively. The corresponding solutions to the Gross–Pitaevskii equation for the condensate wave function are presented in Section 6. In Section 7, they are employed in the analysis of the Bogoliubov transformation and couplings for estimates of the multimode squeezing parameters. We discuss also the multimode dimensionality, squeezing and interference, which determine an asymptotic parameter for computational ♯P-hardness of atomic boson sampling and require analysis of the Bogoliubov–De Gennes equations for the quasiparticle spectrum and eigenfunctions. The experimental aspects of atomic boson sampling are discussed in Section 8. Section 9 contains concluding remarks.

## 2. General Requirements for the BEC Trap Design Facilitating Atomic Boson Sampling

Although any general-case BEC trap can be employed for studying manifestations of its potential quantum advantage over classical computing of boson sampling [1], in order to test this advantage in a controllable and unambiguous way, one should better use a specially designed trapping potential (see an example in Figure 1).

The challenge of the BEC trap design is twofold. On the one hand, it is desirable to have a trap with a finite (or even mesoscopic) number, *M*, of lower split-off excited states or groups of states, which are predominantly populated and strongly coupled to each other by means of Bogoliubov coupling. The atomic boson sampling could refer to the so-called marginal statistics—the quantum statistics of the occupations of these excited states irrespective to the occupations of all other states. It is especially informative if all of the higher excited states are separated from such a split-off miniband or sub-miniband of the selected *M* lower excited states or some groups of them by an energy gap ΔE wider than the temperature *T* and are not significantly coupled to the lower energy states. Then, these higher states are relatively poor populated, do not contribute to ♯P-hard complexity and can be skipped or accounted for as a kind of perturbation.

On the other hand, it is required to provide a way to simultaneously measure, that is to sample, the occupations of those *M* excited particle states or groups of them, say, by means of multi-detector optical imaging based on the light transmission through or scattering from the atomic cloud. Each detector should measure an appropriate occupation by projecting upon a prescribed state or group of states. Moreover, this subset of states or groups of states should be variable and controllable by means of tuning the detectors.

The geometry of a 2D multi-qubit BEC trap, such as the one shown in Figure 1, looks especially convenient for such boson sampling experiments since it allows one to implement multi-detector imaging by means of the laser light passing through the trap perpendicular to its plane. A controllable reconfiguration of the system of detectors aimed at varying the states or groups of states prescribed for occupation sampling also looks easier in the 2D geometry.

Suppose we design a multi-qubit BEC trap with a confining potential U(r) supporting a finite number *Q* of single-qubit cells, which form a 1D, 2D or 3D lattice and have two split-off lower energy levels each. Those two levels appear when a twofold-degenerate ground level is split by a certain perturbation. Then, such a lattice of single-qubit cells should be placed on top of a slightly varying in space background potential with high walls at the trap borders, and the inter-cell potential walls should be adjusted to be relatively high but narrow enough to allow for a quantum tunneling of atoms between cells. All this is necessary for establishment of the common to all cells nonuniform condensate and significant interaction between atoms from different single-qubit cells. The last two conditions are required for the existence of significant Bogoliubov coupling between a large number of excited states without which the multimode squeezing as well as interference via dressed quasiparticles are not well pronounced in the Bogoliubov transformation matrix *R* in Equation (Equation 2) and, hence, the computational ♯P-hardness disappears.

In fact, building a confining potential in the form of a single-qubit cell and duplicating it into a lattice is a relatively straightforward enterprise since such potentials are reminiscent of a double-well potential and an optical lattice potential, in which the BEC as well as the Bogoliubov excitations had been studied a lot [35,43,44,75,76,77,78,79,80,81]. The size of the multi-qubit trap depends on its dimensionality. In the 2D case of Figure 1, an overall dimension of the BEC trap is about $\sqrt{Q}\;\mu$m, since each single-qubit cell has a scale of a de Broglie wavelength ∼1 μm.

The starting point of our analysis is the limiting case of infinitely high inter-cell barriers and identical single-qubit cells, each with two single-particle eigenfunctions corresponding to the first and second energy levels, $e_{1}$ and $e_{2}$. These eigenfunctions form a natural basis for constructing the single-particle excited states of the actual trap. There are $2^{Q}$ combinations of these single-qubit states which are the eigenfunctions of the whole multi-qubit trap. Their Q+1 different energy levels $\left\{\varepsilon_{q}=\left(Q-q\right)e_{1}+qe_{2};\;q=0,1,\ldots ,Q\right\}$ constitute a lower energy miniband. Degeneracies of levels are given by binomial coefficients gq=Qq. Their sum coincides with the number of the single-qubit-state combinations: ∑q=0QQq=2Q.

Those limiting-case eigenfunctions of the multi-qubit trap emerge adiabatically from the wavefunctions of an empty flat box trap when the inter-cell potential barriers are gradually introduced. This process can be easily understood within a 1D model of a flat background potential and almost equally spaced delta-function potential barriers. Its analysis shows that the limiting-case eigenfunctions of the finite lattice of independent qubit cells correspond to some superpositions of the 2Q lower-energy eigenfunctions of the whole trap with finite barriers. Hence, the 2Q lower-energy eigenfunctions $\psi_{n}\equiv \psi_{s,p}$ of order n=p+sQ,p=1,…,Q,s=0,1, of the actual trap with finite barriers can be considered as a system of the “generating” eigenfunctions constituting two bands (s=0,1) and enumerated by the intra-band index p=1,…,Q and the band index s=0,1,….

Although one could consider all $2^{Q}$ energy levels associated with *Q* qubits, it is more convenient to operate with a smaller number of the single-particle energy levels M+1 which constitute some miniband. In Section 4 and Section 5, we show that it is possible to choose the trap parameters in such a way that M+1=2Q levels will form a lower miniband separated from the higher energy levels by an energy gap ΔE wider than the temperature *T*. Below, we mainly discuss the multi-qubit trap properties associated with such a miniband.

Analysis of the case when each cell has a larger number, d>2, of the lower split-off energy levels is very similar. Then, a finite lattice of such cells forms a multi-qudit trap.

In any case, the eigenfunctions $\psi_{s,p}$ and energy levels of the actual trap with a finite trapping potential can be easily controlled and varied in a wide range by means of controlling the background and barrier potentials as well as the dimensions of the single-qubit cells. For instance, the relative occupations of cells, i.e., the relative wavefunction amplitudes in different cells, within an eigenfunction of a given order can be individually controlled by tuning the cell background potentials. The intra-cell qubit properties, including the energy splittings $\delta E_{j},j=1,\ldots ,Q$, can be addressed by adjusting the intra-cell barriers.

The ground-state properties also can be controlled in this way. Implementing also control of the interparticle interactions via the Feshbach resonance [65], one can adjust the condensate wave function as needed. Below, we consider a favorable for the atomic boson sampling regime of a common condensate which is macroscopically occupied and inhomogeneously spread over the entire trap at a low temperature $T\ll T_{c}$ and a relatively large number of trapped atoms N≫Q. At certain conditions, a particular number *M* (for example, M=2Q−1) of lower-miniband excited states can be considered as being decoupled from the continuum of excited states of the total infinite-size Hilbert space and constituting a finite-size Hilbert subspace. The situation could become especially clean and favorable for atomic boson sampling experiments if, in addition, the Bogoliubov couplings are adjusted to be spread over the whole lower miniband but not above the energy gap.

Apparently, the multi-qubit trap is capable of providing a whole series of other BEC regimes [78], starting from a strongly correlated regime and a regime of anomalous fluctuations in the critical region at $T\approx T_{c}$ to the regimes of fragmented condensates of the individual single-qubit cells and a quasi-condensate. However, their discussion goes beyond the scope of the present paper.

## 3. A Basic Model of a Multi-Qubit BEC Trap

The first step in designing the multi-qubit trap is to find its single-particle energy spectrum $\left\{\varepsilon_{n}|n=0,1,\ldots \right\}$ and eigenstates $\left\{\psi_{n}\right\}$, given by the linear Schr$\ddot{o}$dinger equation
(8)$$\left(-\frac{\hbar^{2}}{2m}\Delta +U\left(r\right)\right)\psi_{n}\left(r\right)=\varepsilon_{n}\psi_{n}\left(r\right),$$
and adjust the trap parameters in order to fulfill the requirements on the split-off lower-energy miniband formulated above. The energies may be counted from the energy $\varepsilon_{0}$ of a nondegenerate ground state n=0. An integer *n* orders all eigenstates in increasing energies $\varepsilon_{0}<\varepsilon_{1}\leq \varepsilon_{2}\leq \cdots$ . Solutions to the single-particle Schr$\ddot{o}$dinger Equation (Equation 8) provide a valuable starting point for the design of the multi-qubit trap—the zero-order approximation for the energies and wave functions of the excited states (n=1,2,⋯) as well as the wave function of the ground state (n=0).

The second step is to find how the repulsive interparticle interaction modifies the ground state, that is, to find the condensate wave function $\phi_{0}\left(r\right)$ which obeys the Gross–Pitaevskii equation (the nonlinear Schr$\ddot{o}$dinger equation) [35,52,53]
(9)$$\left(-\frac{\hbar^{2}\Delta }{2m}+U\left(r\right)+gN_{0}{\left|\phi_{0}\left(r\right)\right|}^{2}+2gn_{\mathrm{ex}}\left(r\right)-\mu \right)\phi_{0}=0,\;g=\frac{4\pi \hbar^{2}a}{m}.$$
The goal is to verify the presence of a non-fragmented condensate, which is common for the entire trap and spreading over all single-qubit cells. A non-uniformity of the condensate should be controllable by adjusting the trap parameters. Accurate knowledge of the condensate wave function is necessary for calculating the Bogoliubov couplings
(10)$$\Delta_{kk^{\prime }}=g\;N_{0}\;\int \phi_{k}^{*}\left(r\right)\;{\left|\phi_{0}\left(r\right)\right|}^{2}\;\phi_{k^{\prime }}\left(r\right)\;d^{3}r,\;{\tilde{\Delta }}_{kk^{\prime }}=g\;N_{0}\;\int \phi_{k}^{*}\left(r\right)\;\phi_{0}{\left(r\right)}^{2}\;\phi_{k^{\prime }}^{*}\left(r\right)\;d^{3}r$$
between the preselected bare-particle excited states $\left\{\phi_{k}|\;k=1,2,\ldots \right\}$ and making sure that they are well pronounced for a large enough number of these states as per requirements stated in Section 2. If most of atoms are in the condensate, $N_{0}\approx N$, then a characteristic length of a condensate inhomogeneity is equal to a so-called healing length
(11)$$\xi =\frac{\hbar }{\sqrt{2mgN/V}}=\frac{1}{\sqrt{8\pi aN/V}}\;.$$
The Gross–Pitaevskii equation, as a mean field approximation, is valid if an average distance *d* between atoms is small compared to the healing length,
(12)d≪ξ.

The next step involves solving the Bogoliubov–De Gennes equations for the quasiparticle spectrum and eigenfunctions as well as calculation of the Bogoliubov transformation matrix, squeezing and other parameters describing the joint probability distribution of the excited atom occupations and atomic boson sampling. We just briefly comment on this step in Section 7 and Section 8, since the analysis of quasiparticles goes beyond the scope of this article.

In the present paper, we limit ourselves to the first two steps and calculation of Bogoliubov couplings responsible for interparticle interactions in the Bogoliubov Hamiltonian.

For the sake of clarity and simplicity, we consider only a simple basic model of the multi-qubit BEC trap illustrated in Figure 2: namely, a one-dimensional (1D) or two-dimensional (2D) array of a finite number *Q* of the single-qubit cells. In the case of a 1D chain of the single-qubit cells, each *q*-th single-qubit cell includes two flat background potentials $U_{2q-1},U_{2q}$ and a delta-function potential $\beta_{q}\delta \left(x-x_{q}\right)$ located near its center, while the cells are separated by the delta-function potential walls $\left\{\alpha_{q}\delta \left(x-X_{q}\right)\geq 0|\;q=1,2,\ldots ,Q-1\right\}$ and ordered along the *x* axis so that $0=X_{0}\leq x_{1}\leq X_{1}\leq x_{2}\leq X_{2}\leq \ldots \leq x_{Q}\leq X_{Q}=QL$. The corresponding 1D trapping potential is modeled as follows
(13)$$\begin{array}{cc} \; & U\left(x\right)=\sum_{q=1}^{Q}\{U_{2q-1}\left[\theta \left(x-X_{q-1}\right)-\theta \left(x-x_{q}\right)\right]+U_{2q}\left[\theta \left(x-x_{q}\right)-\theta \left(x-X_{q}\right)\right]\\ \; & \;\;\;+\beta_{q}\delta \left(x-x_{q}\right)+\alpha_{q}\delta \left(x-X_{q}\right)\}\;\mathrm{if}\;x\in \left(0,QL\right);\\ \; & U(x)=\infty \;\mathrm{if}\;x\leq 0\;\mathrm{or}\;x\geq QL. \end{array}$$
The amplitudes of the background potentials $\left\{U_{j}|\;j=1,\ldots ,2Q\right\}$ and all delta-function potentials $\left\{\alpha_{q}\right\},\left\{\beta_{q}\right\}$ as well as their locations $\left\{x_{q}\right\},\left\{X_{q}\right\}$ could be different for different single-qubit cells and constitute a set of controllable parameters of the multi-qubit BEC trap; δ(x) is the Dirac delta function, and θ(x) is the unit step function: θ(x)=0 if x<0, θ(x)=1 if x≥0.

In the case of a 2D square $Q_{1}\times Q_{1}$ array of $Q={\left(Q_{1}\right)}^{2}$ single-qubit cells, we adopt a model potential $U\left(x,y\right)=U\left(x\right)+U^{\prime }\left(y\right)$ given by a sum of two 1D potentials along the axes *x* and *y*, each of which being similar to the 1D potential in Equation (Equation 13):(14)$$\begin{array}{cc} \; & U\left(x\right)=\sum_{q=1}^{Q_{1}}\{U_{2q-1}\left[\theta \left(x-X_{q-1}\right)-\theta \left(x-x_{q}\right)\right]+U_{2q}\left[\theta \left(x-x_{q}\right)-\theta \left(x-X_{q}\right)\right]\\ \; & \;\;\;+\beta_{q}\delta \left(x-x_{q}\right)+\alpha_{q}\delta \left(x-X_{q}\right)\}\;\mathrm{if}\;x\in \left(0,Q_{1}L\right);\\ \; & U^{\prime }\left(y\right)=\sum_{q=1}^{Q_{1}}\{U_{2q-1}^{\prime }\left[\theta \left(y-Y_{q-1}\right)-\theta \left(y-y_{q}\right)\right]+U_{2q}^{\prime }\left[\theta \left(y-y_{q}\right)-\theta \left(y-Y_{q}\right)\right]\\ \; & \;\;\;+\beta_{q}^{\prime }\delta \left(y-y_{q}\right)+\alpha_{q}^{\prime }\delta \left(y-Y_{q}\right)\}\;\mathrm{if}\;y\in \left(0,Q_{1}L^{\prime }\right). \end{array}$$
Again, we set the potential to be infinitely high beyond the outer borders of the entire multi-qubit trap: $U\left(x\right)=\infty \;\mathrm{if}\;x\leq 0\;\mathrm{or}\;x\geq Q_{1}L,\;U^{\prime }\left(y\right)=\infty \;\mathrm{if}\;y\leq 0\;\mathrm{or}\;y\geq Q_{1}L^{\prime }$. The amplitudes of the background potentials $\left\{U_{j}|\;j=1,\ldots ,2Q\right\},\left\{U_{j}^{\prime }|\;j=1,\ldots ,2Q\right\}$ and all delta-function potentials $\left\{\alpha_{q}\right\},\left\{\beta_{q}\right\},\left\{\alpha_{q}^{\prime }\right\},\left\{\beta_{q}^{\prime }\right\}$ as well as their locations $\left\{x_{q}\right\},\left\{X_{q}\right\},\left\{y_{q}\right\},\left\{Y_{q}\right\}$ constitute a set of controllable parameters of the 2D multi-qubit BEC trap.

Modeling the confining potential by piecewise flat and delta-function potentials is a well-justified textbook approach pertinent to the analysis of the effects of tunneling, reflection and trapping of particles by potential barriers and walls on the wave functions and energy spectrum in quantum mechanics (see, e.g., [82,83,84,85] and references therein). It is consistent with the well-known facts that (a) the Rayleigh–Ritz characterization of the eigen energies involves only weighted averages of the potential and (b) the multiple-scale perturbation theory yields the correct leading-order asymptotics within the piecewise-flat-potentials approximation [86]. The main quantities in question for the analysis in the present paper are the condensate wave function and Bogoliubov couplings, which determine the ultimate result for the covariance matrix and statistics of atomic boson sampling. Their representativeness and robustness with respect to the adopted modeling by the piecewise flat and delta-function potentials are predetermined by the nature of the Bogoliubov couplings (Equation 10) as the overlapping integrals which do not depend significantly on a jump in the value of the first or second derivative of the wave function originated from the presence of the delta- or step-function, respectively, in the external potential. Furthermore, the actual potential in the Gross–Pitaevskii and Bogoliubov–de Gennes Equations (Equation 9) and (Equation 34) is always curved by the interparticle–interaction contribution $gN_{0}\phi_{0}^{2}\left(r\right)$ proportional to the continuous condensate occupation $|\phi_{0}^{2}\left(r\right)|$. Obviously, in an experimental setting, a non-flat background potential will lead to qualitatively the same results.

## 4. One-Dimensional Multi-Qubit Trap: Single-Particle Eigen Functions and Energies

Consider a 1D trap with the model potential (Equation 13). The basic model adopted above allows one to solve the 1D Schr$\ddot{o}$dinger Equation (Equation 8),
(15)$$\left(-\frac{\hbar^{2}}{2m}\frac{d^{2}}{d^{2}x}+U\left(x\right)\right)\psi_{n}\left(x\right)=\varepsilon_{n}\psi_{n}\left(x\right),$$
analytically and easily find the single-particle energy spectrum and wave eigenfunctions. In this section, we demonstrate the single-particle properties of the 1D multi-qubit traps in a series of generic examples.

### 4.1. Asymmetric 1D Single-Qubit Trap: Explicit Solution for a Double-Well Trap

The solution to Equation (Equation 15) for the eigen functions and energies of an asymmetric 1D single-qubit trap described by the model (Equation 13) of a double-well trap with the intra-cell delta-function potential of a magnitude β located at a position $x_{1}=\eta L,\;\eta \in \left(0,1\right)$, is elementary:(16)$$\psi_{n}\left(x\right)=A\sin\left(k_{n}x\right)\;\mathrm{if}\;0\leq x\leq \eta L,\;\psi_{n}\left(x\right)=\frac{A\sin(\eta k_{n}L)}{\sin[\left(1-\eta \right)k_{n}L]}\sin\left[k_{n}\left(L-x\right)\right]\;\mathrm{if}\;\eta L\leq x\leq L,$$
(17)$$\sin\left(\eta k_{n}L\right)\sin\left[\left(1-\eta \right)k_{n}L\right]=-\frac{\hbar^{2}k_{n}}{2m\beta }\sin\left(k_{n}L\right),\;\varepsilon_{n}=\frac{\hbar^{2}k_{n}^{2}}{2m}.$$
Here, *A* is an appropriate normalization constant. The dependence of the first six eigen wave numbers $k_{n}$ on the asymmetry parameter η∈(0,1) is illustrated in Figure 3. Note a very narrow energy splitting $\varepsilon_{2}-\varepsilon_{1}\ll \varepsilon_{2}$ between the two lower excited states n=1,2 and a very wide energy gap $\varepsilon_{3}-\varepsilon_{2}\approx 3\varepsilon_{2}$ separating them from the next two excited states n=3,4 in the case of the central, symmetric location of the intra-cell delta-function potential, η=1/2. With an increasing asymmetry, the energy ladder experiences a significant restructuring. For example, if the asymmetry is η≈1/3 or 2/3, then already, the three lower (the lowest first, $\varepsilon_{1}$, and two very close second and third, $\varepsilon_{2}\approx \varepsilon_{3}$) energy levels are separated from the higher energy levels by an energy gap $\varepsilon_{4}-\varepsilon_{3}\approx 1.3\varepsilon_{3}$.

### 4.2. Symmetric 1D Two-Qubit Trap: Even versus Odd Eigenfunctions and Their Eigenenergies

Consider the symmetric two-qubit trap (Equation 13) with the central locations of the intra-cell delta-function potentials of equal magnitude $\beta_{1}=\beta_{2}\equiv \beta$ at $x_{1}=L/2$, $x_{2}=3L/2$ and the inter-cell delta-function potential of the magnitude $\alpha_{1}\equiv \alpha$ at $X_{1}=L$ in the absence of the background potential, $U_{1}=U_{2}=0$. The odd wave functions, which have the odd spatial symmetry relative to the center of the trap, equal zero at the trap center and are not affected by the inter-cell potential wall. Obviously, solutions for them are reduced to the single-qubit trap solution (Equation 16) in each of two single-qubit cells. For example, the odd wave function in the left single-qubit cell is
(18)$$\psi_{n}\left(x\right)=A\sin\left(k_{n}x\right)\;\;\mathrm{if}\;\;0\leq x\leq L/2,\;\psi_{n}\left(x\right)=A\sin\left[k_{n}\left(L-x\right)\right]\;\;\mathrm{if}\;\;L/2\leq x\leq L.$$
Hence, the dimensionless energy spectrum, ${\bar{\varepsilon }}_{n}=\left(2mL^{2}/\hbar^{2}\right)\varepsilon_{n}$, of the odd eigenfunctions is given by Equation (Equation 17), that is
(19)$$k_{n}L\cos\left(k_{n}L/2\right)+\bar{\beta }\sin\left(k_{n}L/2\right)=0,\;{\bar{\varepsilon }}_{n}={\left(k_{n}L\right)}^{2};\;\;\bar{\alpha }=\frac{\alpha mL}{\hbar^{2}},\;\bar{\beta }=\frac{\beta mL}{\hbar^{2}}.$$
A dimensionless parameter $\bar{\beta }$ describes the effect of the intra-cell delta-function potential.

The solution to Equation (Equation 15) for the even eigenfunctions is more involved:(20)$$\begin{array}{cc} \; & \psi_{n}\left(x\right)=A_{1}\sin\left(k_{n}x\right)\;\mathrm{if}\;0\leq x\leq L/2,\\ \; & \psi_{n}\left(x\right)=A_{2}\sin\left[k_{n}\left(L-x\right)+\varphi \right]\;\mathrm{if}\;L/2\leq x\leq L,\\ \; & \psi_{n}\left(x\right)=A_{2}\sin\left[k_{n}\left(x-L\right)+\varphi \right]\;\mathrm{if}\;L\leq x\leq 3L/2,\\ \; & \psi_{n}\left(x\right)=A_{1}\sin\left[k_{n}\left(2L-x\right)\right]\;\mathrm{if}\;3L/2\leq x\leq 2L. \end{array}$$
It includes two normalization constants $A_{1},A_{2}$, the phase shift φ (such that $\tan\varphi =k_{n}L/\bar{\alpha }$) and depends on the inter-cell delta-function potential via the dimensionless parameter $\bar{\alpha }=\alpha mL/\hbar^{2}$. The energy spectrum ${\bar{\varepsilon }}_{n}={\left(k_{n}L\right)}^{2}$ of the even eigenfunctions is determined by the eigen wave number $k_{n}$ that can be found from the explicit transcendental equation:(21)$$k_{n}L\left[k_{n}L\cos\left(k_{n}L\right)+\bar{\beta }\sin\left(k_{n}L\right)\right]+2\bar{\alpha }\sin\left(\frac{k_{n}L}{2}\right)\left[k_{n}L\cos\left(\frac{k_{n}L}{2}\right)+\bar{\beta }\sin\left(\frac{k_{n}L}{2}\right)\right]=0.$$

Figure 4 shows clearly the full structure of energy spectrum of a two-qubit trap. Firstly, one can see the unperturbed energy level spread for an empty rectangular well when the inter-cell and intra-cell potentials are zero. This behavior can be modulated in two ways, by increasing either of the two potentials. There is almost a complete symmetry between how these two potentials effect the energy level structure, with the only difference coming in the even-numbered energy levels. While every fourth level is totally unperturbed by both potentials, the other even energy levels only see the intra-cell potentials, as these functions are zero at the center of the well. This leads to an asymmetry in the structure, which affects the orange-colored energy levels in the figure.

In addition to this asymmetry, the overall structure of the two-qubit-trap energy spectrum is largely determined by the formation of minibands. If either of the two potentials are individually raised to be large, four sub-minibands are formed, while the raising of both potentials leads to the formation of two minibands, with a large energy gap between the first four and second four energy levels. This can be quantitatively measured by taking the ratio of the energy separation between the first and second miniband with the energy width of the first miniband (see Figure 5). This will demonstrate how easy it will be to set the temperature such that the lower miniband is fully populated while the higher has little to no occupation.This figure of merit must be balanced against the necessity that atoms are still able to easily move between cells, requiring that the potential barriers not be too high.

### 4.3. Four-Qubit Chain of Identical Symmetric Single-Qubit Cells: Hierarchy of Even/Odd Solutions

The analysis presented in Section 4.2 can be easily generalized to the case of the four-qubit trap (Equation 13) with similarly symmetric parameters $\alpha_{q}=\alpha ,\beta_{q}=\beta ,x_{q}=\left(q-1/2\right)L,$$X_{q}=qL,U_{2q-1}=U_{2q}=0$∀q=1,2,3,4 and total length 4L. Again, the solutions for the wave functions with the odd spatial symmetry relative to the center of the trap and their energy spectrum are reduced to the solutions for the half trap, that is, for the two-qubit trap and, hence, are given (say, for the left half of the four-qubit trap) by Equations (Equation 18)–(Equation 21). The only novel element of the analysis is the solution for the even eigenfunctions. It has the following form in the left half of the four-qubit trap
(22)$$\begin{array}{cc} \; & \psi_{n}\left(x\right)=A_{1}\sin\left(k_{n}x\right)\;\mathrm{if}\;0\leq x\leq L/2,\\ \; & \psi_{n}\left(x\right)=A_{2}\sin\left[k_{n}\left(L-x\right)+\varphi_{2}\right]\;\mathrm{if}\;L/2\leq x\leq L,\\ \; & \psi_{n}\left(x\right)=A_{3}\sin\left[k_{n}\left(x-L\right)+\varphi_{3}\right]\;\mathrm{if}\;L\leq x\leq 3L/2,\\ \; & \psi_{n}\left(x\right)=A_{4}\sin\left[k_{n}\left(2L-x\right)+\varphi_{4}\right]\;\mathrm{if}\;3L/2\leq x\leq 2L, \end{array}$$
with the same form of equations being found reflected across the center of the trap at x=2L. Now, it includes four normalization constants $A_{1},A_{2},A_{3},A_{4}$, three phase shifts $\varphi_{2},\varphi_{3},\varphi_{4}$ and the eigen wave number $k_{n}$. The latter four quantities can be found from the following four equations expressing a discontinuity of the wave-function derivative across each delta-function potential barrier:(23)$$\begin{array}{cc} \; & k_{n}L\left[\cot\left(\varphi_{4}\right)+\cot\left(\varphi_{4}\right)\right]=2\bar{\alpha },\\ \; & k_{n}L\left[\cot\left(\varphi_{3}+k_{n}L/2\right)+\cot\left(\varphi_{4}+k_{n}L/2\right)\right]=-2\bar{\beta },\\ \; & k_{n}L\left[\cot\left(\varphi_{2}\right)+\cot\left(\varphi_{3}\right)\right]=2\bar{\alpha },\\ \; & k_{n}L\left[\cot\left(k_{n}L/2\right)+\cot\left(k_{n}L/2+\varphi_{2}\right)\right]=-2\bar{\beta }. \end{array}$$
Excluding the phase shifts, we arrive to the explicit transcendental equation,
(24)$$\begin{array}{l} {\left(k_{n}L/2\right)}^{3}\left[\bar{\beta }\sin\left(2k_{n}L\right)+k_{n}L\cos\left(2k_{n}L\right)\right]\\ \;\;\;\;\;\;+{\left(k_{n}L/2\right)}^{2}\left(2\bar{\alpha }+\bar{\beta }\right)\sin\left(k_{n}L\right)\left[\bar{\beta }\sin\left(k_{n}L\right)+k_{n}L\cos\left(k_{n}L\right)\right]\\ \;\;\;\;\;\;\;+{\bar{\alpha }}^{2}\sin^{2}\left(k_{n}L/2\right){\left[\bar{\beta }\sin\left(k_{n}L/2\right)+k_{n}L\sin\left(k_{n}L/2\right)\right]}^{2}\\ \;\;\;\;\;\;\;\;\;\;\;+k_{n}L\bar{\alpha }\bar{\beta }\sin^{2}\left(k_{n}L/2\right)\left[\bar{\beta }\sin\left(k_{n}L\right)+k_{n}L\cos\left(k_{n}L\right)\right]=0, \end{array}$$
for the eigen wave number $k_{n}$ which determines the dimensionless energy spectrum ${\bar{\varepsilon }}_{n}={\left(k_{n}L\right)}^{2}$ of the even eigenfunctions.

The entire energy spectrum is illustrated in Figure 6 by dependence of the first sixteen lower energy levels on the inter-cell and intra-cell delta-function potentials. As expected, it is similar to the analogous dependence for the two-qubit trap shown in Figure 4. Again, on the far left, the unperturbed energy levels can be observed and compared to the energy levels on the far right, which show clearly the expected miniband behavior, with a larger gap between the strongly grouped first eight and second eight energy levels.

Let us look again at the ratio of the energy gap between the first and second minibands and the energy width of the first miniband in Figure 5. Although this ratio is smaller for similar delta-function potentials of the two-qubit trap, the doubling of the available energy levels is a strong advantage. The degradation is not significant, as to once again achieve a ratio of about 3.5, we only need to go from a dimensionless delta-function potential magnitude of 10 to about 12.

Apparently, the even/odd hierarchy of solutions revealed above is suggestive for an extension to any 1D chain of $Q=2^{p},p=1,2,3,\ldots$, identical symmetric single-qubit cells.

### 4.4. Multi-Qubit Chain of Q Identical Single-Qubit Cells: Asymptotics of Zeroes and Miniband of 2Q Energy Levels

Consider a 1D chain of *Q* identical symmetric single-qubit cells, each with a zero flat potential and of length *L*, separated by delta-function potential walls of the same amplitude $\alpha_{q}=\alpha$. The system is placed inside an infinitely high box potential well of length QL. Assume that each single-qubit cell contains a delta-function potential of the equal amplitude, $\beta_{q}=\beta$, placed at the center of the cell and perturbing its energy levels.

As was explained in Section 2, the eigenfunctions of the multi-qubit trap {$\psi_{s,p}\left(x\right)|\;p=1,\cdots ,Q;s=0,1,\cdots$} can be considered as arising adiabatically with increasing delta-function potentials α and β from the sinusoidal wave eigenfunctions of a box trap with a zero flat potential and of length QL,
(25)$$\psi_{n}^{\left(0\right)}\left(x\right)=\sqrt{\frac{2}{QL}}\sin\left(\frac{n\pi x}{QL}\right),\;\varepsilon_{n}^{\left(0\right)}=\frac{{\left(\hbar \pi n\right)}^{2}}{2m{\left(QL\right)}^{2}},\;n=p+sQ;\;p=1,\cdots ,Q;\;s=0,1,\cdots .$$
The index *n* is equal to the number of half-wavelength variations between the borders of the entire box trap and orders the wave functions in accord with the linearly growing numbers of zeroes, n−1, and quadratically growing energies, $\varepsilon_{n}^{\left(0\right)}$. The index s=0,1,⋯ enumerates the bands. Each band consists of *Q* eigenfunctions enumerated by the intra-band index p=1,⋯,Q.

We find a general asymptotic rule: When $\alpha_{q}\to \infty$, all *Q* eigenfunctions $\psi_{s,p}$ within a given *s*-band have exactly the same (equal to the band order *s*) number of zeroes inside each single-qubit cell. The only exception constitutes such single-qubit cells and such eigenfunctions $\psi_{s,p}$ for which there is just one zero of the corresponding sinusoidal eigenfunction $\psi_{p+sQ}^{\left(0\right)}$ located exactly at the center of a single-qubit cell. This is an exceptional, degenerate case of a node frustration when neither of the two delta-function walls of the single-qubit cell are able to shift the location of this zero toward (underneath) its (wall’s) location with increasing delta-potential $\alpha_{q}\to \infty$. The amplitude of this eigenfunction tends to zero everywhere inside such an exceptional single-qubit cell.

The remarkable asymptotic behavior stated above is a consequence of the fact that the eigenfunctions $\psi_{s,p}\left(x\right)$ tend to zero at the positions of the inter-cell delta-potential walls with increasing magnitude of the delta-function potential: $\psi_{s,p}\left(x=jL\right)\to 0\;\mathrm{at}\;\alpha_{q}\to \infty$ for j=1,…,Q−1. This occurs via two mechanisms. A delta-potential wall either (i) gradually digs a deep dip forcing the eigenfunctions to approach zero at the wall location, or (ii) gradually shifts the closest-to-the-wall zero of the sinusoidal wave eigenfunction $\psi_{p+sQ}^{\left(0\right)}$ to (underneath) the wall location. Accordingly, the eigenfunctions do not or do change their sign across the delta-potential wall. It is illustrated in Figure 7(left), where both mechanisms of asymptotics formation are clearly represented. In particular, the inter-cell delta-potential walls at the dimensionless positions x/L=2 and x/L=6 implement the first mechanism on the eigenfunctions $\psi_{s,p=1}$ (blue) and $\psi_{s,p=2}$ (yellow), the second mechanism on the eigenfunction $\psi_{s,p=3}$ (green), and do not affect the eigenfunction $\psi_{s,p=4}$ (red) whose sinusoidal counterpart $\psi_{4+sQ}^{\left(0\right)}$ is already equal to zero at the wall locations. Another situation when the above two mechanisms clearly manifest themselves is discussed in the next Section 4.5 in regard to Figure 11.

The sinusoidal wave functions $\psi_{n}^{\left(0\right)}$ of the higher band orders s≥2, i.e., n>2Q, have three or more zeroes at least within one, say *j*-th, single-qubit cell, that is, within the interval x∈[qL,(q+1)L]. So, at $\alpha_{q}\to \infty \;\forall q$, they turn into the eigenfunctions $\psi_{s,p},s\geq 2$, which has the *s* (two or more) zeroes inside each single-qubit cell and cannot be associated with the $2^{Q}$ eigenfunctions of the multi-qubit lowest miniband that have no more than one zero inside a single-qubit cell. Hence, only the first two bands of the eigenfunctions, $\psi_{0,p}$ and $\psi_{1,p}$, are relevant to the wave function superpositions that asymptotically yield the $2^{Q}$ eigenfunctions of the multi-qubit trap (i.e., combinations of the single-qubit states with the energies within the miniband).

Thus, we focus below on the analysis of the miniband of the first 2Q energy levels corresponding to the first two bands, s=0,1, of the eigenfunctions $\psi_{s,p}$. In principle, we can build a new system of *Q* qubits assigning an arbitrary pair of eigenfunctions $\psi_{0,p}$ and $\psi_{1,p^{\prime }}$ to be the lower and upper energy states of any new qubit. The most natural system of *Q* qubits will be formed by the pairs of eigenfunctions $\left\{\psi_{0,p},\psi_{1,p}|p=1,\ldots ,Q\right\}$ with equal indices $p^{\prime }=p$. Then, we again can consider their $2^{Q}$ multi-qubit combinations of the eigenfunctions of the miniband of the lowest 2Q energy levels in the multi-qubit trap with arbitrary finite (not necessary infinite) inter-cell potential walls. These qubits are not identical anymore, even if their lengths $L_{q},q=1,\ldots ,Q$ are the same.

### 4.5. Multi-Qubit Chain of Significantly Different Single-Qubit Cells: Control of Occupations, Energies

If the single-qubit cells in the multi-qubit chain are not identical, then the excited-state wave functions become less symmetric. However, by controlling the background flat potentials $\left\{U_{j}|\;j=1,\ldots ,2Q\right\}$ in Equation (Equation 13), one can make the ground state more uniform. A typical example of spatial profiles of the ground-state wave function and three lower-energy excited-state eigenfunctions is shown in Figure 8. In this figure and throughout the present paper as in Equation (Equation 19), a bar above a symbol of the potential or other energy quantity denotes its dimensionless value in terms of the energy unit $\hbar^{2}/\left(2mL^{2}\right)$ where *L* is the length of a typical single-qubit cell, that is,
(26)$${\bar{U}}_{j}=\left(2mL^{2}/\hbar^{2}\right)U_{j},\;{\bar{\varepsilon }}_{n}=\left(2mL^{2}/\hbar^{2}\right)\varepsilon_{n}\;.$$

Relatively large variations in the lengths, $L_{q}$, and background flat potentials, $U_{2q-1},U_{2q}$, of different single-qubit cells allow one to control and vary the trap eigenfunctions $\psi_{s,p}$ and energy levels in a wide range. Adjusting separately the flat potentials $U_{j}$ in different single-qubit cells allows one to control individually the relative amplitudes of eigenfunctions in different cells, that is, in particular, the relative occupations of different single-qubit cells as is illustrated in the right column of Figure 7.

The general structure and asymptotic behavior of the trap eigenfunctions described above for the chain of identical single-qubit cells is robust (remains qualitatively the same) with respect to small variations of the trap parameters. However, large variations change the picture. In particular, the single-qubit cells of significantly different lengths could acquire different numbers of eigenfunction zeroes per a single-qubit cell even within the same band of eigenfunctions as is illustrated in Figure 9a.

At last, tuning the intra-cell delta-function potentials $\beta_{q},q=1,\ldots ,Q$, provides one more tool for controlling and varying the profile and energy spectrum of the multi-qubit trap eigenfunctions $\psi_{s,p}\left(x\right)$. As is illustrated in Figure 9b, it affects the eigenfunctions in the first, s=0, band stronger than the eigenfunctions in the second, s=1, band. Thus, it is an efficient tool for controlling the intra-qubit properties, in particular, the qubit energy splittings $\delta E_{q},q=1,\ldots ,Q$.

One other possibility when constructing these multi-qubit traps is to place the inter-cell and intra-cell delta potentials in such a way as to break the symmetry between each qubit cell. This can act as yet another knob by which to control the diversity of the system along with the heights of the delta potentials and the modulation of the background potential. Delta-function potentials can be moved individually or following some group pattern. The position of the potential can be represented by a number from 0 to 1, essentially what percentage of the cell is traveled starting from the center of the cell before dropping down the delta-function potential. Suppose the intra-cell barriers are shifted leftwards on the left side of the four-qubit trap and an equal distance rightward on the right side. The effect of this shifting on the energy levels of a four-qubit trap can be seen in Figure 10. It is not symmetric and can be understood by observing where the delta-function potentials are modulating the unperturbed wave functions for each energy level. For example, take the ninth energy level seen in cyan in Figure 10. This energy level is maximized around 0.18 and minimized around 0.36. When the positions of the delta-function potentials are overlaid on the unperturbed ninth eigenfunction, it becomes clear why this is the case. At the maximum effect, the delta-function potentials are positioned near the maxima of the unperturbed function, while at the minimum effect, the delta potentials are placed instead near the zeros of the unperturbed function, as seen in Figure 11. The latter figure illustrates also the two mechanisms of a wave-function pertubation stated in Section 4.4. In the former case, the first mechanism, namely, digging a dip in a function profile underneath the delta-function potential, takes place. In the latter case, the second mechanism, namely, dragging the nearest node underneath the delta-function potential, takes place. The respective effects on the energy of the eigenfunctions are very different as is shown in Figure 11. Note that the digging effect of the central inter-cell delta-function potential is the same in both cases.

## 5. Two-Dimensional Multi-Qubit Trap: Single-Particle Eigen Functions and Energies

Here, we describe a simple 2D model of the multi-qubit trap formed by a potential, $U\left(x,y\right)=U_{x}\left(x\right)+U_{y}\left(y\right)$, which is the sum of the two 1D potentials considered above. In this case, the solution to the 2D single-particle Schr$\ddot{o}$dinger Equation (Equation 8) is reduced, via a factorization, to the solutions to the 1D Schr$\ddot{o}$dinger equation described above.

As a generic example, let us consider the 2D symmetric four-qubit trap, in which both the potential in the x and y plane are exactly the same. Both will share the same inter-cell and intra-cell delta-function potential strengths. As the energy levels are known for the 1D case (see, for example, Figure 6), constructing the energy levels for the 2D case is a simple task, requiring only for the individual energies to be added in every possible combination. A visualization of the energy levels created by combining the first nine energy levels in each dimension is shown in Figure 12 for the symmetric 2D (4×4)-qubit trap. This fully covers the first miniband of each dimension plus the first energy level in the second miniband.

This structure is interesting. Firstly, let us understand why there is a noticeable asymmetry between the effects of the inter-cell and intra-cell potentials. We see that for the two different axes, the first set of energy levels that are bunched together contains either nine levels for intra-cell potentials or 16 levels for inter-cell potentials. This comes from the fact that for the 1D case of a large intra-cell potential with little inter-cell potential, there is a splitting of the energy levels in the first miniband into two parts, one containing the first three with the other containing the remaining five. These two sub-minibands have a significant enough energy gap between them so that when the 2D plot is created, we see the formation of three sub-minibands, one containing combinations consisting of only energy levels in the first half of the sub-miniband, one containing the crossover terms, and the final piece containing the combinations consisting of only energy levels in the second half of the sub-miniband. Because there are nine possible combinations for the first half, we see nine energy levels in the first band of the plot. However, for the case of the large inter-cell potential, we instead see a 4–4 split of the energy levels in the miniband rather than the 3–5 split in the intra-cell case. Thus, 16 possible combinations of the energy levels in the first half of the miniband.

To understand why there is a difference in splitting of the energy levels depending on if the inter-cell or intra-cell potentials dominate the trap, we need only to look at the fourth energy level and how it is affected by each of the two different types of traps. The easiest to understand is the inter-cell potential dominant traps. In this case, the fourth eigenfunction is almost totally unperturbed by the delta-function potentials, as its natural nodes are already placed at the locations of the inter-cell potential barriers, while the lower three energy levels are pulled up toward the fourth. Likewise, the fifth, sixth, and seventh energy levels are bought up to the totally unperturbed eighth, leading to the 4–4 split structure we observe. However, in the case of the intra-cell potentials dominating the trap, the natural nodes of the fourth energy level wavefunction are placed directly between two of the intra-cell potential walls. This, as described in Section 4.4, leads to a situation where new nodes must be created to accommodate the large potentials. These new nodes drastically increase the average derivative, bringing the energy level much further above the third energy level below it, whose eigenfunction is able to shift its nodes to fall under the existing large potentials without much effect to its energy level. This effect is illustrated in Figure 13.

The last notable aspect of Figure 12 is the crossing of energy levels that can be seen near this origin. This behavior arises from the fact that while initially, the energy levels are approximately evenly spread, once the potentials start ramping up, there are some significant gaps created in the miniband structure. Thus, for low potentials, the energy level created by combining the first and fifth energy levels may be lower than that created by the fourth energy level combined with itself. However, once the fifth energy level is drastically raised by the introduction of the potentials, the combination of the first and fifth levels will increase its total energy above that of the double fourth, at least for the inter-cell dominant case where the fourth energy level is mostly unperturbed. This sort of behavior is only seen near the origin where the energy levels change drastically with the introduction of the delta-function potentials, as once the overall structure begins to form, there is not a significant enough change to create more crossings.

One can also plot the occupation probability distribution for a specific single-particle state in a 2D multi-qubit trap. The spatial profile of the occupation distribution for the single-particle ground state is illustrated in Figure 14. This figure also demonstrates an important property of the multi-qubit trap: namely, that the parameters for the background potential and inter-/intra-cell walls can be tuned to achieve a desirable, in particular, relatively uniform distribution of occupation probability over the entire trap.

## 6. Controlling the Condensate in the Multi-Qubit Trap: The Gross–Pitaevskii Equation

In real interacting gases, the result for the macroscopic condensate wave function given by the Gross–Pitaevskii Equation (Equation 9) significantly deviates from the single-particle ground state of the linear Schr$\ddot{o}$dinger Equation (Equation 8). A difference between the Schr$\ddot{o}$dinger and Gross–Pitaevskii equations originates due to a collective effect of interparticle interactions described by the nonlinear self-interaction term $gN_{0}{\left|\phi_{0}\right|}^{2}\phi_{0}$ in Equation (Equation 9). The main features of the condensate are correctly described already in an approximation neglecting the interaction with the noncondensed fraction (the term $2gn_{\mathrm{ex}}\phi_{0}$ in Equation (Equation 9)) and assuming $N_{0}\approx N$. For simplicity’s sake, we adopt the above approximation and limit discussion to 1D and 2D models of the multi-qubit BEC trap.

A 1D model implies a situation when atoms are tightly confined in the transverse to the *x* axis directions, for example, in a cylinder of length *L*, (y,z)-cross-section area $l_{⊥}^{2}$ with a small transverse size $l_{⊥}\ll L$, and volume $V=Ll_{⊥}^{2}$. Then, in view of the normalization condition $\int_{V}\left|\phi_{0}^{2}\right|d^{3}r=1$ and averaging $\phi_{0}^{2}\left(x,y,z\right)$ over the small y,z cross-section, the 3D condensate wave function $\phi_{0}\left(r\right)$ can be efficiently replaced by a 1D function $\phi_{0}\left(x\right)/l_{⊥}$ that corresponds to a rescaled interaction parameter $g_{1}=g/l_{⊥}^{2}$. Note that the mean-field condition (Equation 12) for validity of the 1D model of the Gross–Pitaevskii Equation (Equation 9) remains the same as in the usual 3D case,
(27)$$8\pi a\ll {\left(\frac{V}{N}\right)}^{1/3},\;\frac{\xi }{d}=\frac{1}{\sqrt{8\pi a{\left(N/V\right)}^{1/3}}},\;g=\frac{4\pi \hbar^{2}a}{m}\;\left(3D\;\;\mathrm{mean}-\mathrm{field}\;\;\mathrm{regime}\right),$$
only if the average distance between atoms is $d={\left(Ll_{⊥}^{2}/N\right)}^{1/3}$, i.e., the volume density of atoms in the trap is large enough: $N/\left(Ll_{⊥}^{2}\right)>1/l_{⊥}^{3}$. Otherwise, the average distance between atoms becomes equal to d=L/N and the mean-field validity condition (Equation 12) imposes a requirement on the scattering length
(28)$$8\pi a\ll Nl_{⊥}^{2}/L,\;\frac{\xi }{d}=\sqrt{\frac{Nl_{⊥}^{2}}{8\pi aL}},\;\;g_{1}=\frac{g}{l_{⊥}^{2}}\;\mathrm{if}\;\;Nl_{⊥}<L\;\;\left(1D\;\;\mathrm{mean}-\mathrm{field}\;\;\mathrm{regime}\right),$$
which is getting more stringent with a decreasing number of atoms. It is worth noting that when aN≫L, the system locally retains the original 3D character despite its 1D geometrical appearance, $L\gg l_{⊥}$. Only in the opposite case, when aN≪L, the system approaches the ground state in the transverse directions and enters the so-called 1D mean-field regime (see [35] and references therein). In the low-density limit $8\pi a\gg Nl_{⊥}^{2}/L$, which corresponds to the strong-coupling 1D limit $g_{1}\equiv g/l_{⊥}^{2}\to \infty$ and is opposite to (Equation 28), the mean-field approach fails and the system becomes the so-called Tonks–Girardeau gas of impenetrable bosons.

Similarly, a 2D model implies a situation when atoms are tightly confined just in one, axial direction, say, along the *z*-axis within a small linear dimension $l_{z}$, while the cross-section area of the trap of volume $V=LL^{\prime }l_{z}$ is relatively large, $LL^{\prime }\gg l_{z}^{2}$. In this case, the 3D condensate wave function $\phi_{0}\left(r\right)$ can be efficiently replaced by a 2D function $\phi_{0}\left(x,y\right)/\sqrt{l_{z}}$ that corresponds to a rescaled interaction parameter $g_{2}=g/l_{z}$. The mean-field condition (Equation 12) for validity of the 2D model of the Gross–Pitaevskii Equation (Equation 9) retains the usual 3D form (Equation 27) only if the average distance between atoms is $d={\left(LL^{\prime }l_{z}/N\right)}^{1/3}$, i.e., the volume density of atoms in the trap is large enough: $N/\left(LL^{\prime }l_{z}\right)\gg 1/l_{z}^{3}$. Otherwise, that is when $Nl_{z}^{2}<LL^{\prime }$, the average distance between atoms becomes equal to $d=\sqrt{LL^{\prime }/N}$ and the mean-field validity condition (Equation 12) is reduced to the requirement on the scattering length
(29)$$8\pi a\ll l_{z},\;\frac{\xi }{d}=\sqrt{\frac{l_{z}}{8\pi a}},\;g_{2}=\frac{g}{l_{z}}\;\mathrm{if}\;Nl_{z}^{2}<LL^{\prime }\;\left(2D\;\;\mathrm{mean}-\mathrm{field}\;\;\mathrm{regime}\right),$$
which is independent on the number of trapped atoms. Again, at extremely low densities, when $|\ln\left(Nl_{z}^{2}/LL^{\prime }\right)|>l_{z}/a$, the mean-field approach fails, the interaction constant $g_{2}=g/l_{z}$ should be replaced by the density-dependent parameter ${\tilde{g}}_{2}=4\pi \hbar^{2}/[m|\ln\left(Nl_{z}^{2}/LL^{\prime }\right)\left|\right]$, and the system enters the regime analogous to the Tonks–Girardeau 1D regime [35].

### 6.1. Single-Qubit Trap: 1D Analytical and 2D Numerical Solutions to the Gross–Piraevskii Equation

The solution to the 1D nonlinear Schr$\ddot{o}$dinger, that is Gross–Piraevskii, Equation (Equation 9) in the stated simple model can be found similar to the solution to the 1D linear Schr$\ddot{o}$dinger Equation (Equation 15), described above, if one employs the elliptic Jacobi function sn(x|p) instead of the exponential function exp(x). For simplicity’s sake, we adopt the Bogoliubov approximation at very low temperature T→0 assuming that practically all atoms are condensed, $N_{0}\approx N$, and the effect of the noncondensed atoms on the condensate is negligible.

Then, the condensate wave function in a box trap with zero potential, U(x)=0, and Dirichlet (zero) boundary conditions is given by the elliptic Jacobi function,
(30)$$\phi_{0}\left(x\right)=\sqrt{\frac{pK(p)}{K(p)-E(p)}}\;\frac{\mathrm{sn}\left(\left.2K\left(p\right)\frac{x}{L}\right|p\right)}{\sqrt{L}},\;\frac{L}{\xi }=\sqrt{8K\left(p\right)\left(K(p)-E(p)\right)\;}.$$
It varies from the half-period sine to an almost constant function (quickly decreasing to zero just in the narrow boundary regions) with the interaction *g* increasing from zero to the larger values. The characteristic scale of the condensate is determined by the healing length (Equation 11). The solution includes complete elliptic integrals of the first and second kinds:(31)$$K\left(p\right)=\int_{0}^{\pi /2}{\left(1-p\sin^{2}\theta \right)}^{-1/2}d\theta ,\;E\left(p\right)=\int_{0}^{\pi /2}{\left(1-p\sin^{2}\theta \right)}^{1/2}d\theta ,$$
According to Equation (Equation 30), the range of the parameter *p* is from 0 to 1. The chemical potential is determined by the normalization condition $\int |\phi_{0}{\left(x\right)|}^{2}dx=1$ as follows
(32)$$\mu =4\left(1+p\right)K^{2}\left(p\right)\hbar^{2}/\left(2mL^{2}\right).$$

The analytical solution in Equation (Equation 30) fully describes the effect of the interparticle interaction on the condensate profile in each single-qubit cell if the inter-cell potential walls are infinitely high, the background potentials are the same in both halves of each cell and the intra-cell delta-function potentials are absent. It is illustrated in Figure 15 for the single-qubit cell. When the healing length is much longer than the cell’s length, ξ≫L, that is, the interaction is very weak and the gas is almost ideal, the parameter *p* is very close to zero. As a result, the condensate profile in each cell is very close to the ground-state solution to the single-particle Schr$\ddot{o}$dinger Equation (Equation 15), that is, a half of the sine function, $\phi_{0}\left(x\right)=\sqrt{2/L}\;\sin\left(\pi x/L\right)$, and $\mu \approx \pi^{2}\hbar^{2}/\left(2mL^{2}\right)$. In the opposite case of a very short healing length, ξ≪L, the parameter *p* approaches 1 and the strong interparticle interaction makes the condensate profile more flat and spread over the entire cell, except for narrow boundary layers of thickness ξ near the walls. Obviously, a similar situation takes place in each half of the single-qubit cells if the intra-cell potential walls are also infinitely high.

The effect of the repulsive interparticle interaction on the condensate profile in the single-qubit box cell in 2D is shown in Figure 16. In the case of an ideal gas, the atoms condense into the ground state $\psi_{0}\left(x,y\right)=\frac{2}{L}\sin\frac{\pi x}{L}\times \sin\frac{\pi y}{L}$ of the single-particle Schr$\ddot{o}$dinger Equation (Equation 15), as shown in Figure 16a. In the case of an interacting gas, the condensate profile $\phi_{0}\left(x,y\right)$ is given by the numerical solution to the 2D Gross–Pitaevskii Equation (Equation 9), as shown in Figure 16b. Comparison of the two plots clearly shows that the particle repulsion flattens the peak of the ground-state wave function and forces the condensate to spread over the entire single-qubit cell. Just the boundary layers of a healing-length thickness remain unoccupied by the condensate.

These effects can be approximately described analytically by a product of the exact analytical solution (Equation 30) to the Gross–Pitaevskii equation in the 1D box trap along the *x*-axis and the similar solution along the *y*-axis,
(33)$$\phi_{0}\left(x,y\right)\approx \frac{pK(p)}{[K(p)-E(p)]L}\;\mathrm{sn}\left(\left.2K\left(p\right)\frac{x}{L}\right|p\right)\mathrm{sn}\left(\left.2K\left(p\right)\frac{y}{L}\right|p\right).$$
Such a 1D-factorization approximation is shown in Figure 16c. It takes into account the interparticle interaction only partially via its separate manifestations along the *x* and *y* dimensions. Comparing Figure 16b and Figure 16c, we conclude that the above approximation slightly underestimates the effect of 2D nonlinear diffusion of the condensate due to the self-interaction $gN_{0}{\left|\phi_{0}\left(x,y\right)\right|}^{2}\phi_{0}\left(x,y\right)$, which is opposite to the phenomenon of self-focusing of an intensive laser light beam in a nonlinear medium. Nevertheless, the 1D-factorization approximation represents the effect of the interparticle interaction on the condensate profile in a box trap in a qualitatively correct fashion.

### 6.2. Condensate Wave Function vs. Single-Particle Ground State in a Multi-Qubit Trap

For a nontrivial multi-qubit BEC trap, due to the presence of the trapping potential U(x)≠0, as shown in Equation (Equation 13), the Gross–Pitaevskii Equation (Equation 9), that is, the nonlinear Schr$\ddot{o}$dinger equation, needs to be solved numerically, for instance, by the method of an imaginary-time evolution (see, e.g., [87]). It is illustrated in Figure 17 and Figure 18 for the case of a four-qubit 1D and 2D trap, respectively.

As a result of the interparticle repulsion, the condensate tends to spread more uniformly over all single-qubit cells. This tendency works against condensate fragmentation [78] and in favor of the formation of a common condensate occupying the entire BEC trap. Moreover, with increasing interaction, the regions of low condensate occupation near the inter- and intra-cell potential walls begin shrinking as well. Both of the above effects significantly increase the number of Bogoliubov-coupled excited states and magnitude of their Bogoliubov couplings in Equation (Equation 10) that favors manifestation of the ♯P-hardness of the atomic boson sampling as is explained in Section 2.

It is worth noting that such a considerable expansion of the condensate shown in Figure 17 and Figure 18 is provided by means of the interparticle interaction alone, without employment of the background potential, which also allows one to control the condensate profile in a similar direction via restructuring the ground-state wave function as is shown in Figure 8 and Figure 14.

Moreover, if the background potential makes the trap asymmetrical, the increasing repulsive interaction tends to restore the trap’s symmetry by converting an asymmetrical single-particle ground state into a more symmetrical condensate wave function. Such evolution of the condensate in the asymmetrical trap is illustrated in Figure 17b and should be compared against the condensate evolution in the symmetrical trap shown in Figure 17a. Clearly, a strong interparticle interaction makes the condensate profiles in both traps almost indistinguishable, while the profiles of the ground state in these traps in the absence of interaction are very different.

## 7. Controlling Multimode Squeezing of Bogoliubov Transform via Bogoliubov Couplings

In equilibrium, the statistics of the many-body system of atoms in the BEC trap is determined by the independent fluctuations of quasiparticles which form the eigenstates of the Bogoliubov Hamiltonian with the eigenenergies $\left\{E_{j}\right\}$ and have the Bose–Einstein occupation number statistics with an average occupation number ${\bar{n}}_{j}={\left(e^{(E_{j}-\mu )/T}-1\right)}^{-1}$. The two-component quasiparticle wave function $\left\{u_{j},v_{j}\right\}$ determine the excited-particle field operator (Equation 3) and obeys the Bogoliubov–de Gennes equations:(34)$$\begin{array}{cc} \; & \hat{L}u_{j}+gN_{0}\;\phi_{0}^{2}\left(r\right)\;v_{j}=+E_{j}u_{j},\\ \; & \hat{L}v_{j}+gN_{0}{\left(\phi_{0}^{*}\right)}^{2}\left(r\right)u_{j}=-E_{j}v_{j}, \end{array}$$
where
$$\hat{L}\equiv -\frac{\hbar^{2}\Delta }{2m}+U\left(r\right)+2g\left(N_{0}{\left|\phi_{0}\left(r\right)\right|}^{2}+n_{\mathrm{ex}}\left(r\right)\right)-\mu ;\;\;n_{\mathrm{ex}}=\sum_{j}\left[|v_{j}{\left(r\right)|}^{2}+\frac{|u_{j}{\left(r\right)|}^{2}+{\left|v_{j}\left(r\right)\right|}^{2}}{\exp(E_{j}/T)-1}\right].$$
In essence, the Bogoliubov–de Gennes equations express the fact of diagonalization of the Hamiltonian (Equation 4) by the Bogoliubov transformation to quasiparticle creation/annihilation operators, as stated in the matrix form in Equation (Equation 6), in the form of differential equations for the coefficients $\left\{u_{j},v_{j}\right\}$ in the expansion of the field operator (Equation 3) via the quasiparticle operators. The wave functions are normalized to unity: $\int_{V}{\left|\phi_{0}\right|}^{2}d^{3}r=1$, $\int_{V}\left(|u_{j}{|}^{2}-{\left|v_{j}\right|}^{2}\right)d^{3}r=1$; j=1,2,…. For simplicity’s sake, hereinafter, we assume that all wave functions $\phi_{0},u_{j},v_{j}$ are real valued. Below, we again neglect by temperature-dependent, Popov’s corrections, that is, skip the contribution due to the noncondensate density $n_{\mathrm{ex}}$ and assume $N_{0}\approx N$.

The matrix *R* of the Bogoliubov transformation (Equation 2) can be found from the equation
(35)$$\left(\begin{array}{c} {\hat{a}}^{†}\\ \hat{a} \end{array}\right)=R\left(\begin{array}{c} {\hat{b}}^{†}\\ \hat{b} \end{array}\right)=\left[\begin{array}{cc} A & B^{*}\\ B & A^{*} \end{array}\right]\left(\begin{array}{c} {\hat{b}}^{†}\\ \hat{b} \end{array}\right)$$
that relates creation and annihilation operators ${\hat{a}}^{†},\hat{a}$ describing bare particles to the operators ${\hat{b}}^{†},\hat{b}$ describing quasiparticles as per Equation (Equation 3). Since the Bogoliubov transformation (Equation 2) leaves the canonical Bose commutation relations invariant, it obeys the symplectic property
(36)$$R\Omega R^{T}=\Omega ,\;\Omega =\left[\begin{array}{cc} 0 & 1\\ -1 & 0 \end{array}\right].$$
Another, equivalent to (Equation 35), representation of the Bogoliubov transformation (Equation 2) can be written in terms of the wave functions, rather than the operators, determining the particle field operator in Equation (Equation 3):(37)$$\left(\begin{array}{c} \phi \\ 0 \end{array}\right)=R\left(\begin{array}{c} u\\ -v^{*} \end{array}\right)\equiv \left[\begin{array}{cc} A & B^{*}\\ B & A^{*} \end{array}\right]\left(\begin{array}{c} u\\ -v^{*} \end{array}\right),$$
(38)$$R^{T}\left(\begin{array}{c} 0\\ \phi \end{array}\right)\equiv \left[\begin{array}{cc} A^{T} & B^{T}\\ B^{†} & A^{†} \end{array}\right]\left(\begin{array}{c} 0\\ \phi \end{array}\right)=\left(\begin{array}{c} \;v^{*}\\ u \end{array}\right).$$
The column-vectors $\phi ={\left(\phi_{1},\phi_{2},\ldots \right)}^{T}$ and $u={\left(u_{1},u_{2},\ldots \right)}^{T}$, $v={\left(v_{1},v_{2},\ldots \right)}^{T}$ are composed of the excited states $\left\{\phi_{k}\right\}$ and quasiparticle wave functions $\left\{u_{j}\right\}$, $\left\{v_{j}\right\}$, respectively.

Projecting Equation (Equation 38) onto a set of the orthonormal excited states $\left\{\phi_{k}|\;k=1,2,\ldots \right\}$ which are also orthogonal to the condensate wave function $\phi_{0}$, we obtain the explicit formulae for the entries of the Bogoliubov block matrices $A=(A_{kj})$ and $B=(B_{kj})$,
(39)$$A_{kj}=\int u_{j}^{*}\left(r\right)\phi_{k}\left(r\right)d^{3}r,\;B_{kj}=\int v_{j}^{*}\left(r\right)\;\phi_{k}^{*}\left(r\right)\;d^{3}r,$$
as overlapping integrals between those bare-particle wave functions and the quasiparticle wave functions given by the solution to the Bogoliubov–De Gennes equations (Equation 34).

The Bogoliubov matrix *R* can be expressed explicitly also via the Bogoliubov couplings in Equation (Equation 10) by means of a pure algebraic diagonalization of the Bogoliubov Hamiltonian in the sense of the matrix Equation (Equation 6). Indeed, in any basis $\left\{\phi_{k}\;|k=1,2,\ldots \right\}$ of excited states, orthogonal to the condensate wave function and constituting the excitation field operator ${\hat{\psi }}_{\mathrm{ex}}=\sum_{k\neq 0}\phi_{k}{\hat{a}}_{k}$ as in Equation (Equation 3), the blocks of the Hamiltonian matrix in Equation (Equation 5) are explicitly given by the Bogoliubov couplings in Equation (Equation 10) as follows
(40)$$K=\left(\varepsilon_{kk^{\prime }}-\mu \delta_{k,k^{\prime }}+2\Delta_{kk^{\prime }}\right),\;\tilde{K}=\frac{1}{2}\left({\tilde{\Delta }}_{kk^{\prime }}\right),\;H=\left[\begin{array}{cc} \tilde{K} & K\\ K^{*} & {\tilde{K}}^{*} \end{array}\right].$$
Here, $\varepsilon_{kk^{\prime }}=\langle \phi_{k}|\hat{\varepsilon }|\phi_{k^{\prime }}\rangle$ is the matrix of the single-particle energy operator $\hat{\varepsilon }=-\hbar^{2}\Delta /\left(2m\right)+U\left(r\right)$ which constitutes the single-particle Schr$\ddot{o}$dinger Equation (Equation 8). In particular, the basis $\left\{\phi_{k}\;|k=1,2,\ldots \right\}$ can be constructed out of the excited-state eigenfunctions $\left\{\psi_{n}\;|n=1,2,\ldots \right\}$ of the Schr$\ddot{o}$dinger Equation (Equation 8) by means of the standard Gram–Schmidt orthonormalization starting from making these functions orthogonal to the condensate wave function $\phi_{0}$. Then, by means of the symplectic property (Equation 36), Equation (Equation 6) determining the Bogoliubov transformation can be rewritten as the following equation
(41)$$\Omega HR=R\left[\begin{array}{cc} E & 0\\ 0 & -E \end{array}\right],\;E=\mathrm{diag}\left\{E_{j}|\;j=1,2,\ldots \right\}.$$
It states that the *j*-th column of the Bogoliubov matrix, $R_{j}={\left\{A_{1j},A_{2j},\ldots ,B_{1j},B_{2j},\ldots \right\}}^{T}$, is the eigenvector of the matrix ΩH corresponding to the quasiparticle eigenenergy $E_{j}$, that is
(42)$$\Omega H\;R_{j}=E_{j}R_{j}\;,\;\Omega H=\left[\begin{array}{cc} K^{*} & {\tilde{K}}^{*}\\ -\tilde{K} & -K \end{array}\right].$$
(There is also the nonphysical eigenvector counterpart $R_{j}^{\left(-\right)}={\left\{B_{1j}^{*},B_{2j}^{*},\ldots ,A_{1j}^{*},A_{2j}^{*},\ldots \right\}}^{T}$ corresponding to the negative eigenenergy $-E_{j}<0$.)

After calculating the Bogoliubov transformation matrix *R* as per Equation (Equation 6), one can find the multimode squeezing parameters [1,68,69,70,71,72,73] from Equation (Equation 2). Quantum statistics of the many-body fluctuations in a BEC trap and, in particular, the computational complexity of the atomic boson sampling are determined by two fundamental sets of eigenvectors and eigenvalues associated with the diagonalization of (a) the squeezing matrix and (b) the Bogoliubov Hamiltonian, as is explained in Section 1 and Section 2. Both of those sets of eigenvectors and eigenvalues are determined by the Bogoliubov couplings (Equation 10) via the Bogoliubov transformation matrix *R*. Thus, the key problem is to calculate the Bogoliubov couplings and understand how many of them can be essentially nonzero and controllable in a wide range within the multi-qubit BEC trap suggested and described in this paper.

Knowing the condensate wave function from the solution to the Gross–Pitaevskii equation outlined in Section 6 and choosing the bare-particle excited states, for example, as the purely harmonic, sine functions or the solutions to the single-particle Schr$\ddot{o}$dinger Equation (Equation 8) (see Section 4 and Section 5), made orthogonal to the condensate and each other via the standard Gram–Schmidt orthonormalization, it is straightforward to calculate the integrals constituting the Bogoliubov couplings (Equation 10) and analyze their set for the multi-qubit BEC trap. The related numerical results are illustrated for the cases of symmetrical and asymmetrical 1D four-qubit traps in the plots shown in Figure 19 and Figure 20, respectively.

First, comparing Figure 20 against Figure 19 makes it clear that the trap asymmetry greatly enlarges the number of Bogoliubov-coupled bare-particle excited states. Indeed, in the asymmetrical trap, the essentially nonzero couplings spread much further from the main diagonal of the Bogoliubov-coupling matrix $\Delta_{k,k^{\prime }}$ than in the symmetrical trap where only narrow lanes of entries around the main diagonal and anti diagonal are essentially nonzero. In addition, the degeneracy of zero coupling between the bare-particle excited states of exactly odd and even spatial parity in the symmetrical trap (Figure 19), that results in exactly zero values of all entries in each diagonal of an odd number parallel to the main diagonal, is essentially broken in the asymmetrical trap (Figure 20). It is restored only in the limit of a very strong interparticle interaction.

Second, the maximum spread of essentially nonzero couplings occurs at a moderate interparticle interaction $\frac{L}{\xi }∼1$ corresponding to the healing length ξ being on order of the single-qubit cell length *L*. Much stronger interaction, $\frac{L}{\xi }\gg 1$, tends to localize nonzero couplings just onto the main and anti diagonals. Both these effects are seen in each row of plots in Figure 19 and Figure 20 where the interaction strength is increasing from left to right.

Third, changing the bare-particle excited states, chosen for the simultaneous atom-number detecting within the atomic boson sampling, from the set generated by the Gram–Schmidt orthonormalization out of the sine functions (the upper rows in Figure 19 and Figure 20) to the set generated out of the solutions to the single-particle Schr$\ddot{o}$dinger equation (the lower rows in Figure 19 and Figure 20) greatly affects the structure of the Bogoliubov-coupling matrix both in the symmetrical and asymmetrical traps.

All of the above observations confirm that the inference the multi-qubit BEC trap provides is an excellent opportunity for controlling the Bogoliubov couplings and, hence, the multi-mode squeezing and interference of bare-atom excited modes in a very wide range. Obviously, the more chaotic, messy, dense and wide the distribution of the essentially nonzero elements over the Bogoliubov-coupling matrix (Equation 10), the more favorable the set of trap’s parameters and bare-atom excited states chosen for detection of atom numbers for testing manifestations of the computational ♯P-hardness of atomic boson sampling. Among patterns shown in Figure 19 and Figure 20, the one in the center of the lower row in Figure 20 is the most representative picture of such a complexity.

The asymptotic parameter of this complexity is determined by the Bogoliubov transformation via a multimode dimensionality of the subspace of the excited-states involved in the squeezing-matrix eigenvectors with essentially nonzero squeezing parameters (see Equation (Equation 2)) and the Hamiltonian-matrix eigenvectors with low enough eigenenergies corresponding to quasiparticles with essentially nonzero populations (see Equation (Equation 42)). In general, this asymptotic parameter increases as the number of groups of excited states chosen for occupation sampling via multi-detector imaging is growing. However, for a given experimental setup with a BEC trap of a finite size, there is a maximum number *M* of modes/channels started from which a further increase of the number of sampled/detected occupations would not essentially increase the complexity of boson sampling.

## 8. Toward Experiments on Atomic Boson Sampling in a BEC Trap

Suppose one has an appropriate BEC trap. (A possible model/example of such a trap is discussed in the previous sections.) Then, as is explained in Section 1.1 and Section 1.2, the excited atoms, by themselves, naturally fluctuate and stay in the squeezed states inside the trap even at thermal equilibrium due to interactions with each other. This allows one to eliminate any nonequilibrium processes or dynamics, such as a precise time-dependent control of system parameters and gates or any other type of processing usually associated with quantum computers or simulators, as well as the sophisticated external sources of squeezed or single bosons (required for photonic sampling) from the atomic sampling experiments. It remains just to split the noncondensate into fractions based on the groups of excited states and to measure the distribution of atom numbers over the chosen groups of excited states by means of appropriate detectors.

For instance, one can divide the volume of the trap into a system of spatial cells. Another possibility is to separate atoms in accord with their velocities, that is, to deal with the cells in the momentum space. Anyway, the measurement of atom numbers could be completed by means of a multi-detector imaging. In a BEC destruction scheme, one switches off the confining trap and allows the cloud of trapped atoms to expand freely. In this case, following a standard time-of-flight measuring technique, it is required to take a few successive images of the expanding cloud and properly interpret them in terms of kinetic equations for expansion. In this way, different spatial or momentum subsets of atoms could be separated from each other, and sampling of their occupation numbers could be obtained.

The imaging technique implies an illumination of the atomic cloud with a laser pulse and measuring its transmitted or scattered components by multiple detectors. The transmitted signal carries information on the absorption, dispersion and polarization transformation of light caused by an atomic cloud [33,75,88,89,90]. The signal due to scattering and fluorescence [91] could be controlled and structured by employing special external cavities and laser sources that support light modes which mimic the excited states preselected for sampling. The optical imaging for atomic boson sampling has much in common with the experiments on the local atom-number fluctuations in BEC traps [47,48,88,90,91,92,93,94].

The spatial or momentum cells/modes represent groups of excited states selected for detection/sampling. Of course, the excited states can be described/composed via an arbitrary basis in the single-particle Hilbert space. Accordingly, the analytical formulae for their joint occupation probability distribution $\rho (\left\{n_{k}\right\})$ and characteristic function, derived in [1], have a universal form, i.e., are valid for any choice of such a basis. A transition from one basis to another one just adds an extra unitary transformation (Equation 7) to the Bogoliubov matrix *R* in Equation (Equation 2). Moreover, the universality of the general result for the characteristic function obtained in [1] extends to the so-called marginal or coarse-grained statistics of occupations of any groups of excited states, that is, to the occupation statistics evaluated irrespective to the occupations of all other excited states. The corresponding “incomplete” experiments on atomic boson sampling are the ones to be devised and implemented in reality. Obviously, the condensed atoms, which constitute the macroscopic condensate wave function $\phi_{0}\left(r\right)$ orthogonal to the excited states $\phi_{k}\left(r\right)$, should not be countered during the sampling procedure.

A computational complexity of atomic boson sampling depends on the number *M* of groups of excited atomic states which are resolved by the multi-detector imaging and are subject to interference due to mixing through the quasiparticles and to squeezing due to interparticle interaction. This number *M* plays a part of the number of channels in the optical interferometer. A mean occupation of the groups of excited states scales as $(N-\;N_{0})/M$. Obviously, by increasing the total number *N* of atoms loaded in the trap and, therefore, the number $N-N_{0}$ of atoms in the noncondensate, one can make larger and, hence, easier for detection the number of atoms in each of *M* groups of preselected excited states. Note, however, that the asymptotic parameter responsible for the ♯P-hardness of atomic boson sampling is not proportional to any of the numbers $N,N-N_{0}$ or *M*.

Modern technology allows one to measure the number of atoms in a specified volume or subset of atoms with nearly single atom resolution [91,93,94,95]. Yet, achieving the single atom accuracy is not absolutely necessary. In particular, the ♯P-hardness of boson sampling exists even in the case of the threshold detecting scheme in which an outcome of the measurement is just zero or non-zero occupation in each preselected group of excited states [16,24,25].

Finally, an experimental setup should provide the means to reconfigure detectors for projecting upon a vastly varying set of groups of excited states, i.e., to accumulate statistics of joint occupations for numerous different subsets of groups of excited states. Only in this way can the quantum advantage be demonstrated at the most challenging level of the average case.

## 9. Concluding Remarks

We introduce the multi-qubit BEC trap for studying manifestations of the quantum many-body statistical phenomena which are ♯P-hard for computing. In particular, we describe the basic properties of the multi-qubit trap, including the single-particle excited states and their energy spectrum via the single-particle Schr$\ddot{o}$dinger Equation (Equation 8), the condensate wave function versus the single-particle ground state via the Gross–Pitaevskii Equation (Equation 9), and the Bogoliubov couplings (Equation 10) between excited states responsible for the formation of quasiparticles and multimode squeezing via the Bogoliubov–de Gennes equations (Equation 34). It is completed within the 1D and 2D models, as shown in Equations (Equation 13) and (Equation 14).

We show that the multi-qubit BEC trap offers a convenient and thoroughgoing control of the many-body system parameters essential for the interplay between excited states’ interference and squeezing. This interplay can be revealed via an apropriate decomposition of the Bogoliubov-transformation matrix in Equation (Equation 2) and is responsible for the computational ♯P-hardness which is the basis for a potential quantum advantage of atomic boson sampling over classical computing [1].

It would be very interesting to study experimentally various phenomena associated with the atomic boson sampling. The BEC trap is a boson-sampling platform alternative to a photonic interferometer. Both systems provide the output multivariate statistics which shows computational ♯P-hardness associated with the hafnian of complex-valued matrices. The proposed multi-qubit trap design discussed in the present paper allows one to vary those matrices and, hence, the output statistics over a wide range. Thus, the latter, major requirement for testing quantum advantage is fulfilled by the multi-qubit BEC trap. The remarkable fact is that classical computing of the hafnian of even relatively low-dimensional matrices corresponding to the number of sampled modes/channels of the order of M=8×8=64 is already inaccessible to modern supercomputers. Especially promising are boson-sampling experiments with the multi-qubit BEC trap containing a finite number *M* of lower-miniband split-off excited states or groups of them (see Figure 1).

The case of a few single-qubit cells with a relatively small number of sampled occupations M=2,3,4,… promises the discovery of new quantum effects similar and beyond a particle analog of the simple Hong-Ou-Mandel interference effect. It can be accomplished by means of the current magneto-optical trapping and detection technology. The value of such experiments for the comprehension of the fundamental aspects of the many-body quantum systems responsible for their computational ♯P-hardness is difficult to overestimate.

The conclusive experiments with an asymptotically large numbers of single-qubit cells, Q≪1, and sampled excited states or groups of them, M≫1, addressing the computational ♯P-hardness of quantum many-body processes are very challenging. Yet, they seem to be within reach and could hit convincing manifestations of quantum advantage.

## Figures and Tables

**Figure 1 entropy-24-01771-f001:**
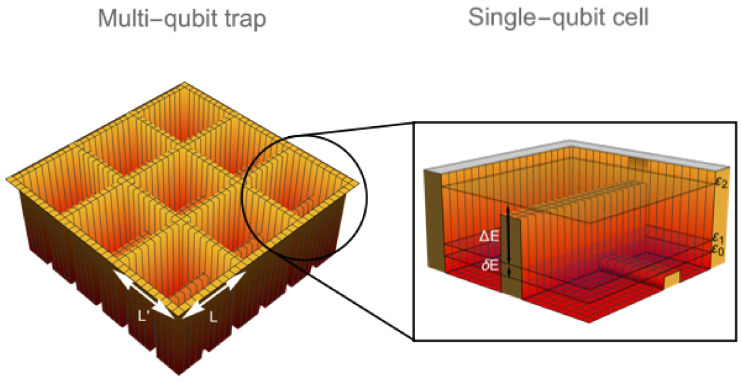
Two-dimensional (2D) model of a BEC trap made up of *Q* single-qubit (or single-qudit) cells of size $L\times L^{\prime }$ each contributing with two (or more) lower energy levels to the lower-energy miniband of the multi-qubit (or multi-qudit) trap. This miniband is separated from the higher energy levels by an energy gap ΔE much larger than the lower-energy splitting δE. For presentation purposes, an inhomogeneous underlying (background) potential, designed for controlling the condensate profile and Bogoliubov couplings, as well as the high potential walls at the outer borders of the trap are not shown.

**Figure 2 entropy-24-01771-f002:**
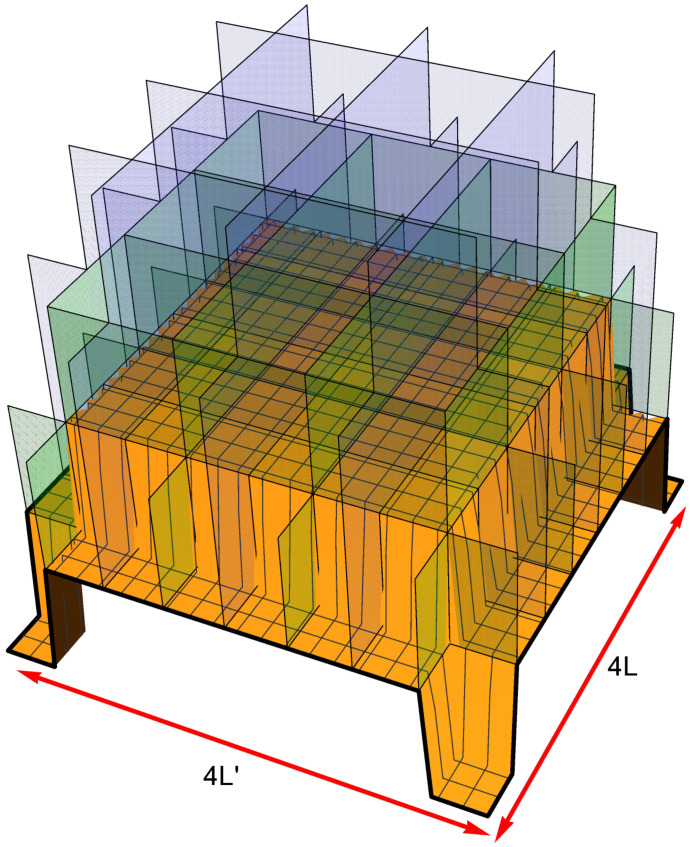
(4×4)-qubit BEC trap of a dimension $4L\times 4L^{\prime }$ as per the 2D model (Equation 14) of trapping potential U(x,y) consisting of the inter- and intra-cell walls atop a central pedestal. The infinite outer walls of the entire box trap are not shown.

**Figure 3 entropy-24-01771-f003:**
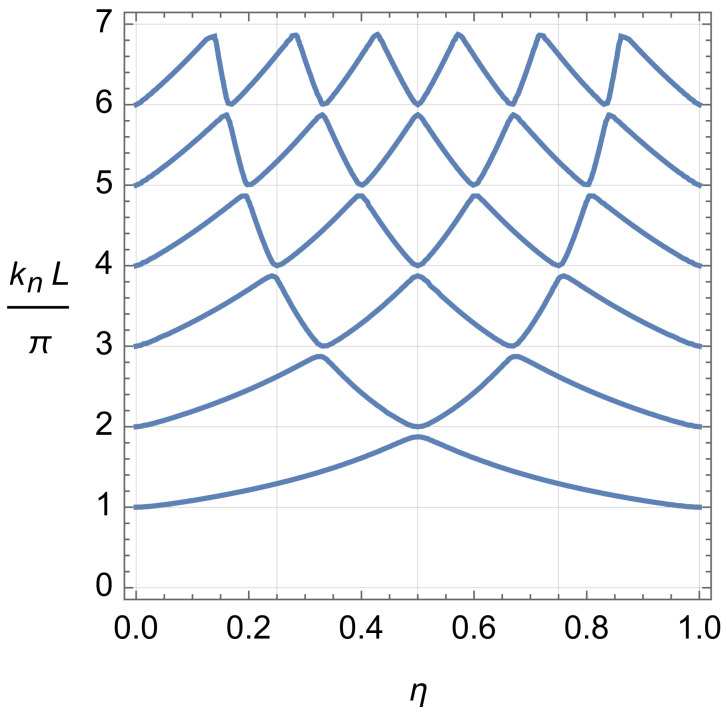
The first six eigen wave numbers $k_{n}$ for the 1D asymmetric single-qubit trap of length *L* as the functions of the position ηL of the intra-cell delta-function potential of the dimensionless amplitude $\beta mL/\hbar^{2}=2.5$.

**Figure 4 entropy-24-01771-f004:**
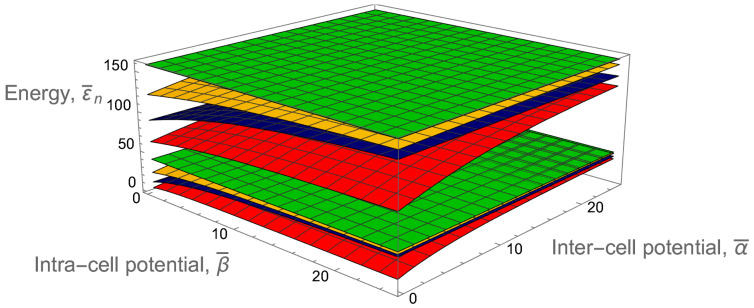
The first eight single-particle energy levels of the 1D symmetric two-qubit trap in the absence of a background potential as they depend on the inter-cell and intra-cell delta-function potential barriers as per Equations (Equation 19) and (Equation 21). On the far left, the unperturbed energy levels can be observed and compared to the energy levels on the far right, which show clearly the miniband behavior, with a larger gap between the strongly grouped first four and second four energy levels.

**Figure 5 entropy-24-01771-f005:**
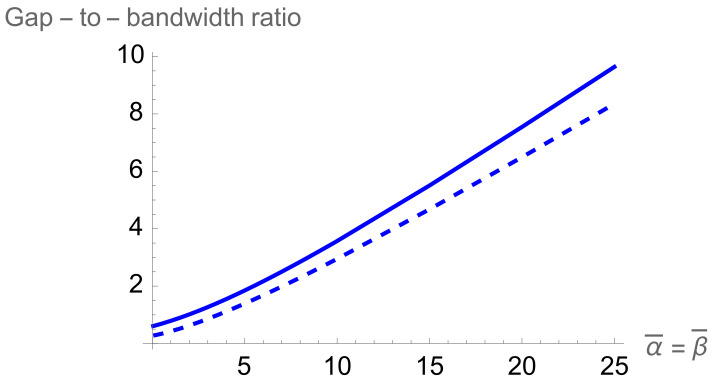
The ratio of the energy gap between the first and second minibands over the width of the first miniband for the symmetric two- (solid) and four- (dashed) qubit traps (Figure 4 and Figure 6) as a function of the equal dimensionless amplitudes of the inter- and intra-cell delta-function potentials, $\bar{\alpha }=\bar{\beta }$.

**Figure 6 entropy-24-01771-f006:**
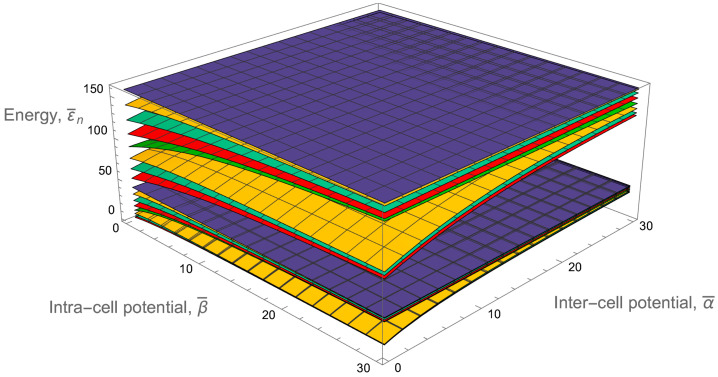
The first sixteen single-particle energy levels of the 1D four-qubit chain of identical symmetric single-qubit cells in the absence of a background potential ($U_{j}=0\;\forall j$) as they depend on the inter-cell and intra-cell delta-function potentials as per Equations (Equation 19), (Equation 21) and (Equation 24).

**Figure 7 entropy-24-01771-f007:**
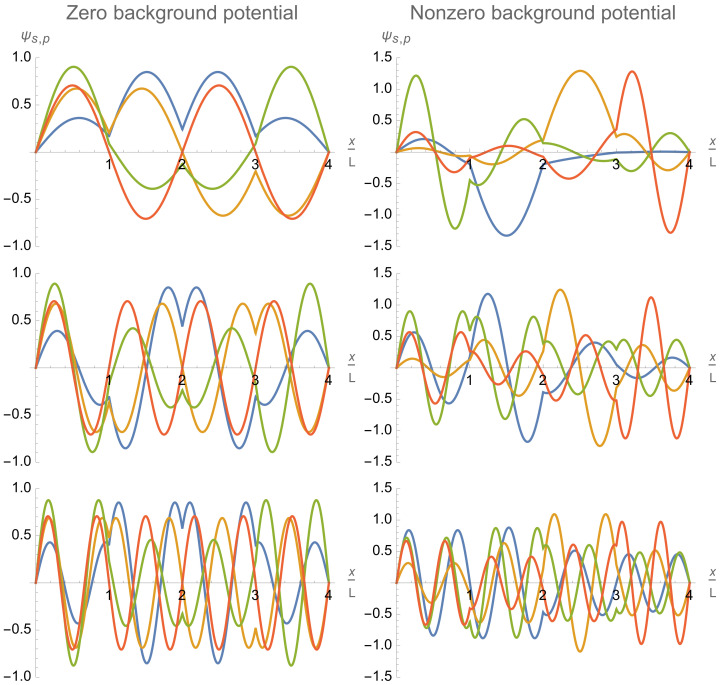
The first three bands, s=0 (1st row), s=1 (2nd row), and s=2 (3rd row), of the eigenfunctions $\psi_{s,p}\left(x\right)$ (p=1 in blue, p=2 in yellow, p=3 in green, p=4 in red) for the 1D chain of Q=4 single-qubit cells. The inter-cell walls are the delta-function potentials of the same magnitude, $\alpha_{q}=9\hbar^{2}/mL$, located at the equally spaced dimensionless positions x/L=1,2,3. There are no intra-cell potentials. The **left** column of graphs: Identical single-qubit cells with zero background potentials, $U_{j}=0$. The **right** column of graphs: The single-qubit cells with different background flat potentials, $U_{j}mL^{2}/\hbar^{2}=0,0,10,10,20,20,0,0$. In both cases, the two mechanisms of the general asymptotic rule (stated in Section 4.4) for a transition from the sinusoidal wave eigenfunctions (Equation 25) of a uniform box trap to the eigenfunctions of a multi-qubit trap are clearly observable.

**Figure 8 entropy-24-01771-f008:**
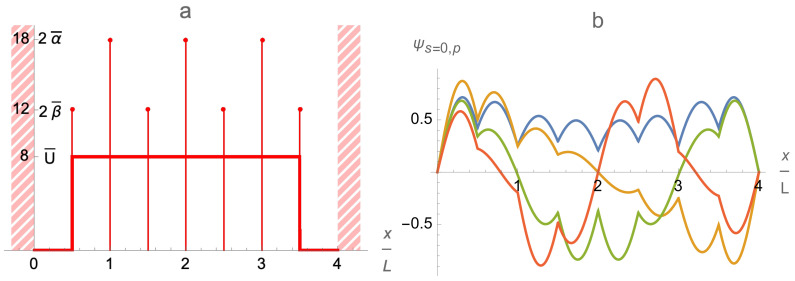
(**a**) An example of a four-qubit trap potential: Three inter-cell and four intra-cell delta-function potential barriers are separating eight flat potential segments in the form of a central pedestal. (**b**) The ground-state (blue) and first three excited-state eigenfunctions for the four-qubit trap (**a**).

**Figure 9 entropy-24-01771-f009:**
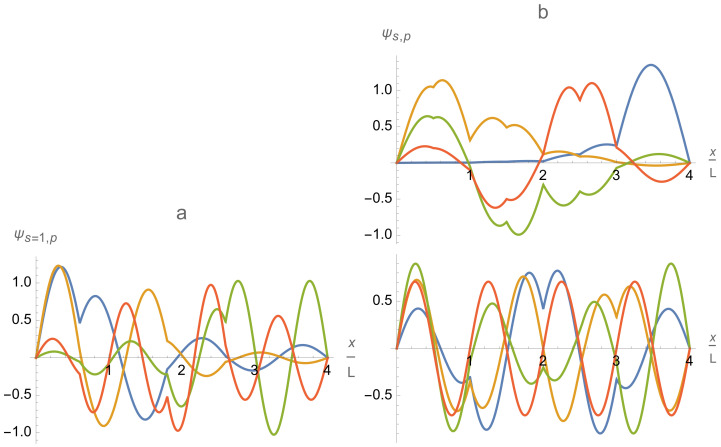
The eigenfunctions $\psi_{s,p}\left(x\right)$ (p=1 in blue, p=2 in yellow, p=3 in green, p=4 in red) for the chain of Q=4 single-qubit cells with a zero background potential, $U_{j}=0,\;j=1,\ldots ,8$. The inter-cell walls are the delta-function potentials of the same magnitude, $\alpha_{q}mL/\hbar^{2}=9,\;q=1,2,3,4$. (**a**) The eigenfunctions $\psi_{s=1,p}\left(x\right)$ of the second band in the case of single-qubit cells of different lengths $L_{q}/L=0.6,1.2,0.8,1.4$. Two mechanisms of the general asymptotic rule (stated in Section 4.4) for a transition from the sinusoidal wave eigenfunctions (Equation 25) of a uniform box trap to the eigenfunctions of a multi-qubit trap are clearly visible. Contrary to the case of identical single-qubit cells in Figure 7(left), now the numbers of zeroes per a single-qubit cell in the eigenfunctions of the second band s=1 are not all equal to unity but could be also zero, two or even three and different in the different cells. (**b**) The eigenfunctions $\psi_{s,p}\left(x\right)$ of the first (s=0) and second (s=1) bands in the case of different delta-function potentials $\beta_{q}mL/\hbar^{2}=1,2,3,0$ at the center of single-qubit cells of equal length *L*. A comparison with the case of identical single-qubit cells in Figure 7(left) shows that now, the eigenfunctions of the band s=0 are notably modified while the eigenfunctions of the band s=1 stay almost intact.

**Figure 10 entropy-24-01771-f010:**
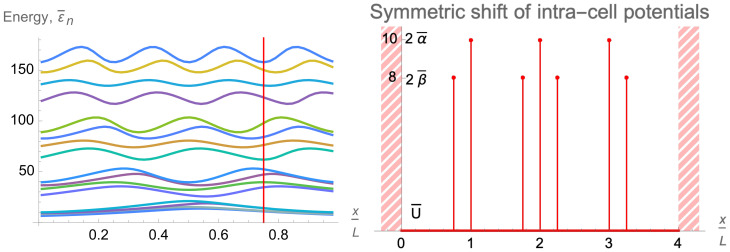
**Left**: The single-particle energy levels for a four-qubit trap with intra-cell potential barriers of strength $\bar{\beta }=4$ and inter-cell barriers of strength $\bar{\alpha }=5$ for various symmetrically shifted positions of the intra-cell potentials. **Right**: The specific trapping potential for the position marked by the red line. Note the symmetry about the center of the trap.

**Figure 11 entropy-24-01771-f011:**
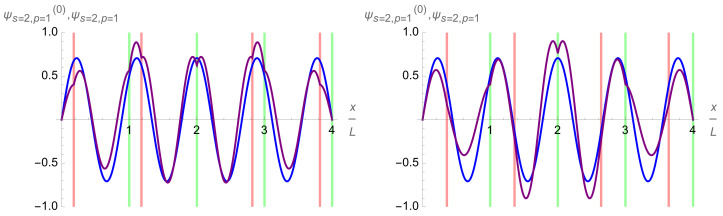
The unperturbed and perturbed ninth-energy-level eigenfunction, $\psi_{s=2,p=1}^{\left(0\right)}$ (in blue) and $\psi_{s=2,p=1}$ (in purple), in the case of the intra-cell delta-function potentials at position 0.18 (**left**) and 0.36 (**right**) marked by red vertical lines. Note how on the right, the intra-cell delta-function potentials act on the eigenfunction near zeros, reducing their effect on the eigenenergy, while on the left, the intra-cell delta-function potentials act on the eigenfunction near two peaks, having a very large impact on the eigenenergy ${\bar{\varepsilon }}_{n=9}$ shown in Figure 10.

**Figure 12 entropy-24-01771-f012:**
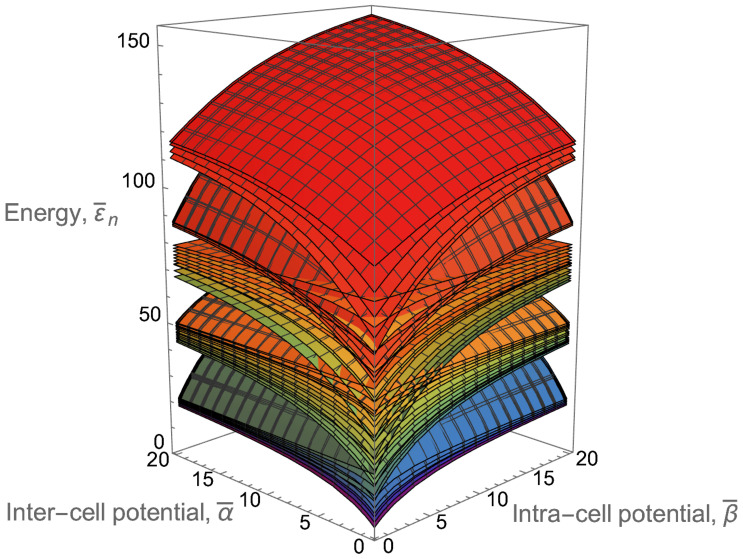
The first 80 energy level combinations comprising the eigenenergies $\left\{{\bar{\varepsilon }}_{n}\right\}$ for the symmetric 2D (4×4)-qubit trap in the absence of a background potential ($U_{j}=0\;\forall j$). Compare this plot against its 1D 4-qubit counterpart in Figure 6 and note how there are energy crossings near the origin of the graph, and that there is a slight asymmetry in the inter-cell and intra-cell directions.

**Figure 13 entropy-24-01771-f013:**
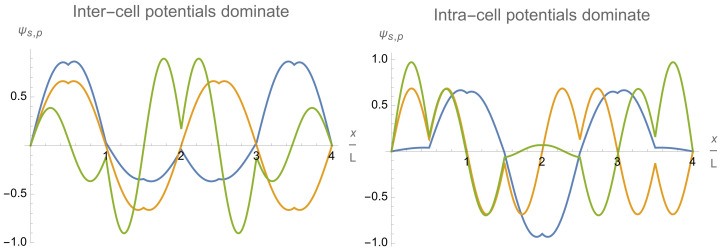
The eigenfunctions $\left\{\psi_{n}\right\}$ for the third (blue), fourth (yellow), and fifth (green) energy levels in the 1D multi-qubit trap of Q=4 identical single-qubit cells for the inter-cell (**left**) and intra-cell (**right**) dominant delta-function potentials $\alpha_{q}=\alpha$ and $\beta_{q}=\beta \;\forall q$. The dominant delta-function potential has a magnitude of $31\hbar^{2}/\left(mL\right)$ while the non-dominant potential has a magnitude of $1\;\hbar^{2}/\left(mL\right)$. Note how in the inter-cell-potential dominant case, the fourth eigenfunction is most similar to the third, while in the intra-cell-potential dominant case, the fourth eigenfunction is most similar to the fifth.

**Figure 14 entropy-24-01771-f014:**
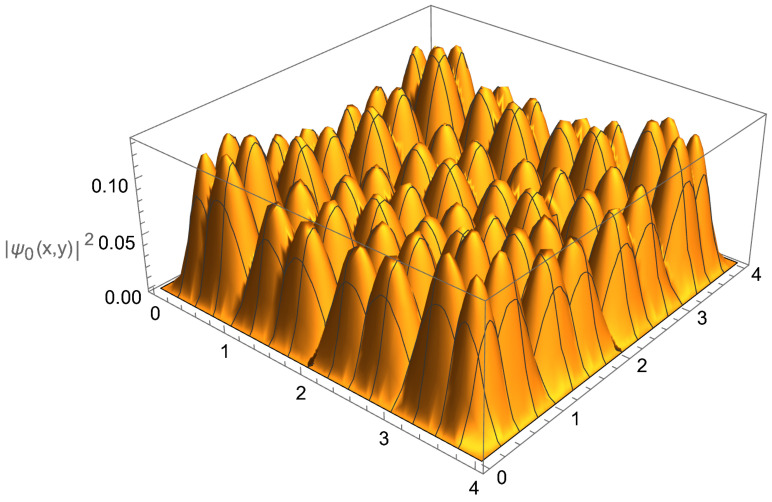
Occupation probability distribution in the single-particle ground state, $|\psi_{0}{\left(x,y\right)|}^{2}$, for the 2D (4×4)-qubit trap with delta-function potentials $\alpha_{q}=4\hbar^{2}/\left(mL\right)$, $\beta_{q}=2\hbar^{2}/\left(mL\right)\;\forall q$ and a central flat pedestal potential $U=8\hbar^{2}/\left(mL^{2}\right)$ ranging from 0.5<x<3.5 and 0.5<y<3.5 as seen in Figure 2.

**Figure 15 entropy-24-01771-f015:**
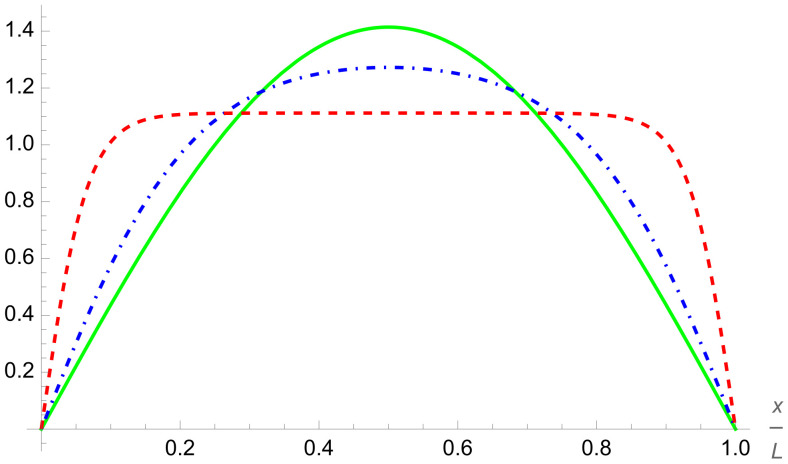
The interparticle interaction makes the condensate more uniform and spread over the entire single-qubit trap as is revealed by the analytical solution (Equation 30) to the Gross–Pitaevskii Equation (Equation 9) in the case of infinitely high inter-cell potential walls and zero background and intra-cell potentials: $\frac{L}{\xi }=0$ (an ideal gas—solid green curve, p=0), $\frac{L}{\xi }=5$ (a moderate interaction—dot-dashed blue curve, p≈0.86), $\frac{L}{\xi }=20$ (a strong interaction—dashed red curve, p≈0.999996).

**Figure 16 entropy-24-01771-f016:**
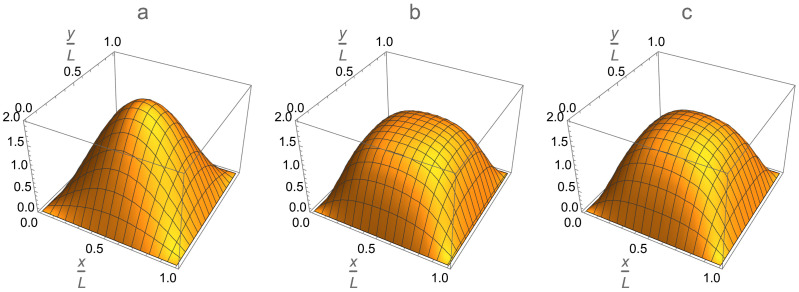
Two-dimensional (2D) single-qubit BEC trap with zero background and intra-cell potentials. The interparticle interaction makes the condensate more uniform and spread over the entire single-qubit cell as is revealed by comparing (**a**) the ground-state wave function $\psi_{0}\left(x,y\right)=\frac{2}{L}\sin\frac{\pi x}{L}\times \sin\frac{\pi y}{L}$ given by the single-particle Schr$\ddot{o}$dinger Equation (Equation 15) in the absence of interaction against (**b**) the condensate wave function $\phi_{0}\left(x,y\right)$ in the presence of interaction, $\frac{L}{\xi }=5$, computed via an exact numerical solution to the Gross–Pitaevskii Equation (Equation 9). The plot (**c**) is an approximation of the latter condensate wave function $\phi_{0}\left(x,y\right)$ via a factorization (Equation 33) of the exact analytical solutions for the 1D box trap, Equation (Equation 30), along the *x* and *y* axes. All three plots present the dimensionless condensate wave function of the unity norm.

**Figure 17 entropy-24-01771-f017:**
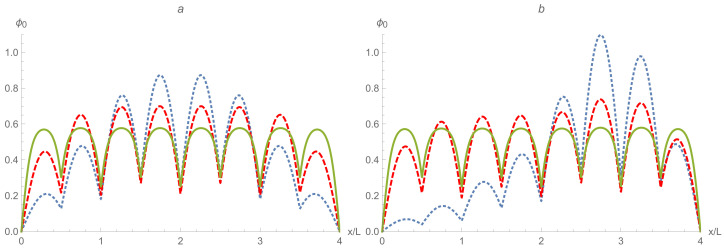
The ground-state wave function according to the single-particle Schr$\ddot{o}$dinger Equation (Equation 15) (blue dotted curve) and the corresponding condensate wave function $\phi_{0}$ according to the Gross–Pitaevskii Equation (Equation 9) in the presence of the moderate, $\frac{L}{\xi }=2$, (red dashed curve) and strong, $\frac{L}{\xi }=10$, (green solid curve) interaction in the case of (**a**) symmetric (U(x)=0) and (**b**) asymmetric ($U\left(x\right)=\left(4\hbar^{2}/\left(mL^{2}\right)\right)\left[\theta \left(x-0.5L\right)-\theta \left(x-2.5L\right)\right]$) 1D four-qubit trap; $\alpha_{j}=1.5\beta_{j}=16\hbar^{2}/\left(mL\right)\;\forall j$.

**Figure 18 entropy-24-01771-f018:**
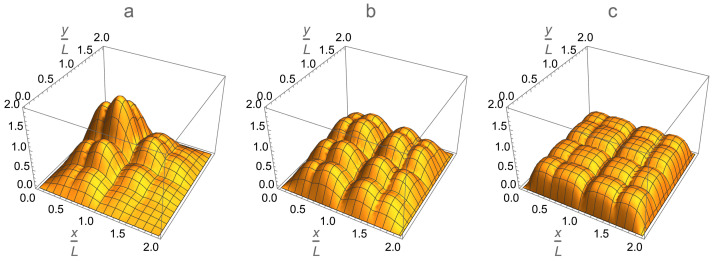
Two-dimensional (2D) (2×2)-qubit BEC trap: (**a**) The ground-state wave function $\psi_{0}$ according to the single-particle Schr$\ddot{o}$dinger Equation (Equation 15) in the absence of interaction as well as the condensate wave function $\phi_{0}$ according to the Gross–Pitaevskii Equation (Equation 9) in the presence of (**b**) moderate, $\frac{L}{\xi }=5$, and (**c**) strong, $\frac{L}{\xi }=20$, interaction; $\alpha_{j}=8\hbar^{2}/\left(mL\right),\alpha_{j}^{\prime }=6\hbar^{2}/\left(mL\right),\beta_{1}=\beta_{1}^{\prime }=4\hbar^{2}/\left(mL\right),\beta_{2}=\beta_{2}^{\prime }=2\hbar^{2}/\left(mL\right),U_{j}=U_{j}^{\prime }=0\;\forall j$ (see Equation (Equation 14)).

**Figure 19 entropy-24-01771-f019:**
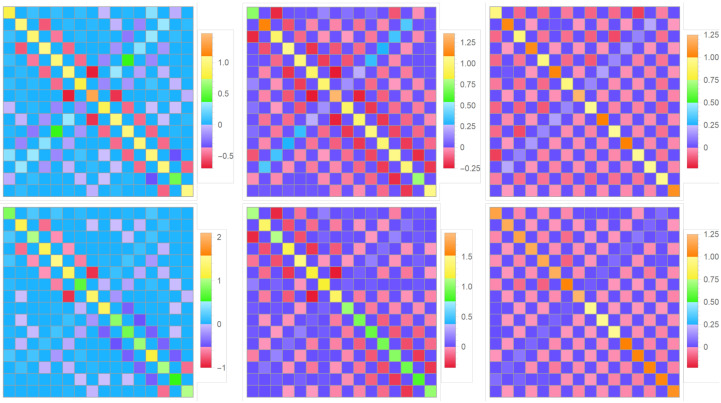
Matrix of Bogoliubov couplings (Equation 10) between the first sixteen excited states in the case of the symmetric 1D four-qubit trap shown in Figure 17a; $\alpha_{j}=1.5\beta_{j}=16\hbar^{2}/\left(mL\right),U_{j}=0\;\forall j$. The excited states are obtained via the Gram–Schmidt orthogonalization from the condensate wave function $\phi_{0}$ and (**upper row**) the sine functions sin(kπx/(4L)),k=1,…,16, or (**lower row**) the first sixteen eigenfunctions of the single-particle Schr$\ddot{o}$dinger Equation (Equation 15) in the presence of (the first column) vanishing, $\frac{L}{\xi }\to 0$, (the second column) moderate, $\frac{L}{\xi }=2$, and (the third column) strong, $\frac{L}{\xi }=10$, interaction.

**Figure 20 entropy-24-01771-f020:**
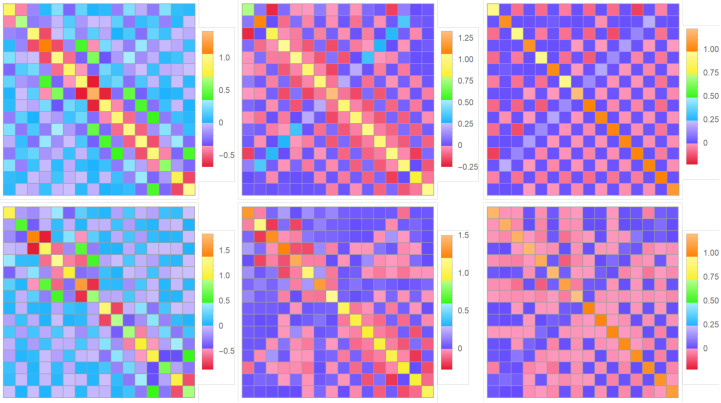
Matrix of Bogoliubov couplings (Equation 10) between the first sixteen excited states in the case of the asymmetric 1D four-qubit trap shown in Figure 17b; $\alpha_{j}=1.5\beta_{j}=16\hbar^{2}/\left(mL\right)\;\forall j,U\left(x\right)=\left(4\hbar^{2}/\left(mL^{2}\right)\right)\left[\theta \left(x-0.5L\right)-\theta \left(x-2.5L\right)\right]$. The excited states are obtained via the Gram–Schmidt orthogonalization from the condensate wave function $\phi_{0}$ and (**upper row**) the sine functions sin(kπx/(4L)),k=1,…,16, or (**lower row**) the first sixteen eigenfunctions of the single-particle Schr$\ddot{o}$dinger Equation (Equation 15) in the presence of (the first column) vanishing, $\frac{L}{\xi }\to 0$, (the second column) moderate, $\frac{L}{\xi }=2$, and (the third column) strong, $\frac{L}{\xi }=10$ interaction.

## Data Availability

Not applicable.

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
