# Peer review of "Multi-Qubit Bose–Einstein Condensate Trap for Atomic Boson Sampling"

_entropy, 2022, doi:10.3390/e24121771_

Round 1

Reviewer 1 Report

This work basically focuses on a systematic numerical analysis of a multi-qubit trap design for Bose-Einstein condensates, consisting of a two-dimensional array of quadrangle infinite boxes or cells divided into two parts by a barrier with the purpose to produce coupled single qubits. Both the inter-cell boundaries and the intra-cell barrier are modeled by Dirac delta-functions (providing a sort of finite two-dimensional Dirac comb type structure), while each side of the cells is described by a constant potential. The latter can be conveniently tuned both between the two sites of a single cell and among all the cells in order to set (quantum) control mechanisms on the condensate qubits. The properties of this potential model are numerically investigated at the level of the linear and nonlinear Schrödinger equations (the latter being modeled by the Gross-Pitaevskii equation), analyzing the consequences of the nonlinearity on the energy levels with respect to the linear regime. Control scenarios and their consequences are also analyzed both in the macroscopic domain (Gross-Pitaevskii equation) and the many-body regime one (Bogoliubov transform).

The work is technically correct, the results are reasonable (although I have not reproduced them, as a computational scientist myself I have not found evidence indicating something wrong or unphysical in them), and, in general, the manuscript is well written, focused on the discussion rather than on formalism. The topic dealt with here can be of some interest to the quantum computing community and hence publishable. However, first the authors should try to reduce the unnecessarily lengthy introductory part (nearly eight pages, including abstract and work summary). In a case like this, particularly if the main ideas and theory are already given somewhere else (for instance, Ref. [1]), the shorter the better. Otherwise, the reader quickly loses track (and interest) after the few first pages. Two or three concise introductory pages would be more than enough, in my opinion, of course, with the first summarizing paragraph taken to the end of the Introduction and not preceding it, as it happens here. As for the rest of the work, although it is also a bit unnecessarily lengthy, still it is affordable because of the intrinsic interest of the discussion of the results and their consequences.

Reviewer 2 Report

In this work, the authors described a multi-qubit BEC trap that can be used for quantum many-body statistical phenomena that are considered #P-hard.  They gave an analysis of the trap properties (energy spectrum, BEC wavefunction solutions, etc.) as an atomic boson sampling platform, specifically in 1D and 2D geometries.  They have shown control mechanisms of this platform for multimode squeezing and excited states coupling. 

Considering the manuscript in its entirety, it was an engaging read and well-written.  I commend the authors for providing a thorough but succinct introduction on atomic boson sampling.  The derivation of the relations used in their analysis of the multi-qubit trap was presented clearly.  The discussion of their results was also well written, and the accompanying figures and plots were sufficient and helpful in the analysis.  

I would recommend acceptance of the manuscript for publication, but I have two minor questions:

1. The model the authors presented for the multi-qubit trap consists of hard-wall/flat potentials.  But in an experimental setting, this is quite difficult to achieve.  How sensitive are the authors’ results to the flatness of the trapping potentials? 

2. In figure 6, in the leftmost corners of the upper six energy level surfaces ((0,0) elements), there are cylindrical columns coming off the surface.  Are these simply artifacts in the rendering of the plot?
